# Mining the equine gut metagenome: poorly-characterized taxa associated with cardiovascular fitness in endurance athletes

Núria Mach [1,2 ✉], Cédric Midoux [3,4,5], Sébastien Leclercq [6], Samuel Pennarun[7], Laurence Le Moyec[8,9], Olivier Rué[3,4], Céline Robert[1,10], Guillaume Sallé[6,11] & Eric Barrey[1,11]

Emerging evidence indicates that the gut microbiome contributes to endurance exercise performance. Still, the extent of its functional and metabolic potential remains unknown. Using elite endurance horses as a model system for exercise responsiveness, we built an integrated horse gut gene catalog comprising ~25 million unique genes and 372 metagenome-assembled genomes. This catalog represents 4179 genera spanning 95 phyla and functional capacities primed to exploit energy from dietary, microbial, and host resources. The holo-omics approach shows that gut microbiomes enriched in *Lachnospiraceae* taxa are negatively associated with cardiovascular capacity. Conversely, more complex and functionally diverse microbiomes are associated with higher glucose concentrations and reduced accumulation of long-chain acylcarnitines and non-esterified fatty acids in plasma, suggesting increased ß-oxidation capacity in the mitochondria. In line with this hypothesis, more fit athletes show upregulation of mitochondrial-related genes involved in energy metabolism, biogenesis, and $Ca^{2+}$ cytosolic transport, all of which are necessary to improve aerobic work power, spare glycogen usage, and enhance cardiovascular capacity. The results identify an associative link between endurance performance and gut microbiome composition and gene function, laying the basis for nutritional interventions that could benefit horse athletes.

[1] Université Paris-Saclay, INRAE, AgroParisTech, GABI, Jouy-en-Josas, France. [2] Université de Toulouse, INRAE, ENVT, IHAP, Toulouse, France. [3] Université Paris-Saclay, INRAE, MaIAGE, Jouy-en-Josas, France. [4] Université Paris-Saclay, INRAE, BioinfOmics, MIGALE bioinformatics facility, Jouy-en-Josas, France. [5] Université Paris-Saclay, INRAE, PROSE, Antony, France. [6] Université François Rabelais de Tours, INRAE, ISP, Nouzilly, France. [7] INRAE, Genomic facility, 31326 Castanet-Tolosan, France. [8] Université d'Évry Val d'Essonne, Université Paris-Saclay, Évry, France. [9] Muséum National d'Histoire Naturelle, CNRS, MCAM, Paris, France. [10] École Nationale Vétérinaire d'Alfort, Maisons-Alfort, France. [11] These authors contributed equally: Guillaume Sallé, Eric Barrey. ✉email: nuria.mach@inrae.fr

Endurance athletes undergo prolonged cardiovascular exercise and withstand physiological stress that disrupts the body's homeostasis. This, in turn, overwhelms organs and the system's normal function[1,2]. The ability to run for long distances at high speed is an uncommon feat for land mammals. Through years of selective breeding for athletic performance and tailored training practices, Arabian horses have gained built-in biological mechanisms to compete at distances of up to 160 km in a single day, an effort comparable to a human marathon or ultramarathon runners[3]. Like humans[4], these equine athletes display well-adapted physiological abilities, including processing large volumes of oxygen required for aerobic metabolism and markedly larger hearts than those less physically active. Unique to horses is their quadrupedal nature of locomotion, the coupling of respiration to stride frequency, and the pronounced increase in pulmonary artery pressure. These equine-specific features place different physiologic loads on the heart during endurance than those observed in humans[4].

Endurance exercise performance primarily depends on cardiovascular fitness, exercise economy, and the ability to sustain both the metabolic and thermoregulatory demands of such activity[4,5] without the accumulation of exponential levels of blood lactate or skeletal muscle fatigue[6–8]. Athletes with the greatest improved cardiovascular fitness and fatigue resistance often succeed in competitions[4,9,10].

Undoubtedly, endurance exercise performance entails complex multifactorial processes whose mechanisms are still not fully understood. New evidence has shown that the gut microbiome and its associated metabolites can act locally in the intestine or accumulate in different body fluids[11] that impact host athletic performance during endurance racing[12,13]. Such gut microbiome-derived metabolites include short-chain fatty acids, dimethyl sulfone, trimethylamine oxide, indoles, tryptamine, oligosaccharides, peptidoglycans, and secondary bile acids[14,15], all of which affect host health[16]. Within the context of exercise, the gut microbiome metabolites support multiple physiological strands, e.g., energy metabolism, hydration, redox reactions, and immune responses, that can affect fatigue and stress perception[2,15,17–20]. For example, *Veillonella atypica* likely enhances athletic endurance performance via the utilization of host lactate and the production of propionate[12,13]. Beyond microbial-derived metabolites, changes in the microbial composition and increased diversity correlate with improved performance and cardiorespiratory fitness in marathon runners regardless of sex, age, body mass index, and diet[21]. Increases of the Firmicutes-Bacteroidetes ratio[22], or depletion in *Eubacterium* spp.[23] enhanced the cardiovascular capacity of athletes, as assessed by the maximum oxygen consumption.

Delineating of the relationships between the gut microbiome and endurance performance is in its infancy in humans, and it is hampered by the appropriate control of known confounding factors (such as diet, training loads, medications, occurring illnesses, environment, and genetic background). In this respect, Arabian horses emerge as a suitable in vivo model for studying the microbiome adaptations in response to endurance exercise. Because of apparent differences in anatomy—e.g., the horse cecum is large relative to the total gastrointestinal tract—and physiology, the relevance, and translation of the observations made in horses to human dietary applications may not be straightforward. However, the natural aptitude of Arabian horses for athletic performance, and the homogeneity of their genetic and environmental backgrounds, offer a more tractable system. The interdependence of exercise performance and gut microbiota in horses is underscored by several lines of evidence[19,20,24–27], although the range and extent of this interplay are mainly unknown. Recent findings suggest that gut microbial metabolites

in endurance horses, e.g., acetate, valerate, dimethyl sulfone, trimethylamine oxide, formate, and secondary bile acids coupled with circulating free fatty acids regulate mitochondrial function by preventing hypoglycemia[25], which is the limiting factor for fatigue onset and, thus, athletic performance. Despite these findings, if and how gut microbiome functions are responsible for better adaptations to resist fatigue and succeed in athletic performances are unresolved.

To address this knowledge gap, we have built an extensive gene catalog of the gut microbiome in elite endurance horses. This expands the current representation of the equine gut microbiome with more than 25 million non-redundant genes identified and 369 new metagenome-assembled genomes (MAGs). Moreover, holo-omics data integration from the host and microbiome domains show that gut microbiome composition, functions, and mitochondria activity are critical determinants for cardiovascular fitness. Relatively poorly-described genera and their pool of genetic resources likely regulate metabolic pathways to fine-tune mitochondrial function and enhance cardiovascular capacity. On the contrary, microbial communities with reduced diversity but a higher abundance of core taxa from the *Lachnospiraceae* family are associated with poorer performance.

## Results

**Building a horse gut microbiome gene catalog and MAG repertoire.** We constructed a microbial gene catalog from the feces of 11 highly trained endurance horses (Supplementary Data 1). After quality filtering and host sequence decontamination, 1124 million high-quality paired reads were available, with an average sequencing depth per sample of 93–107 million paired reads, similar to that used for the construction of the chicken[28] and bovine[29] gut gene catalogs (Supplementary Data 2). These data were de novo assembled (total assembly size of 21.68 Gb) to build a non-redundant gene catalog of 25,250,066 genes with an average length of 618 bp (Supplementary Note, Supplementary Data 2, Supplementary Fig. 1a–e). Individual horses harbored around half of these genes ($n = 11,809,713$; Fig. 1a). The core group of genes present in all individuals consisted of <7.2% of the overall microbial gene pool ($n = 1,805,693$; Fig. 1b). Yet, these core genes showed highly conserved abundance rank structure across individuals, representing 29.7 to 38.4% of the overall microbiome abundance. Only 22.5% ($n = 922,362$) of the recently published horse gut microbiome gene catalog[30] overlapped with our gene catalog, indicating that the functional potential of gut microbiomes in horses is vast and currently under-sampled.

Taxonomic annotation of the microbial gene catalog was implemented with Kaiju[31] using a greedy mode and the NCBI *nr_euk* reference database. While 39% of the gene sequences were unclassified, this approach annotated 61% of the gene sequences (Fig. 1c) and revealed a diverse community of 95 phyla encompassing 1110 families and 4179 unique genera (Supplementary Note, Supplementary Data 3). More than 90% of these genera have not been previously described in horses. Among classified genes, bacteria (95.58% of genes, $n = 2927$ genera) defined most of the assemblage in terms of abundance and diversity, followed by a handful of eukaryotes (2.55% of genes, $n = 1081$ genera), and archaea (1.20% of genes, $n = 170$ genera). At the phylum level, genes related to bacterial phyla Firmicutes (47.1%) and Bacteroidetes (21.8%) greatly outnumbered Proteobacteria (6.0%), Spirochaetes (3.9%), and Actinobacteria (3.1%; Fig. 1d). Consistent with two recent metagenomic studies in horses[30,32], Ascomycota (0.56%) and Basidiomycota (0.25%) were among the top annotated Eukaryota phyla in the gut. At the same time, *Saccharomycetaceae* represented the dominant eukaryote family[30]. Many genes in the catalog pertained to the Ciliophora

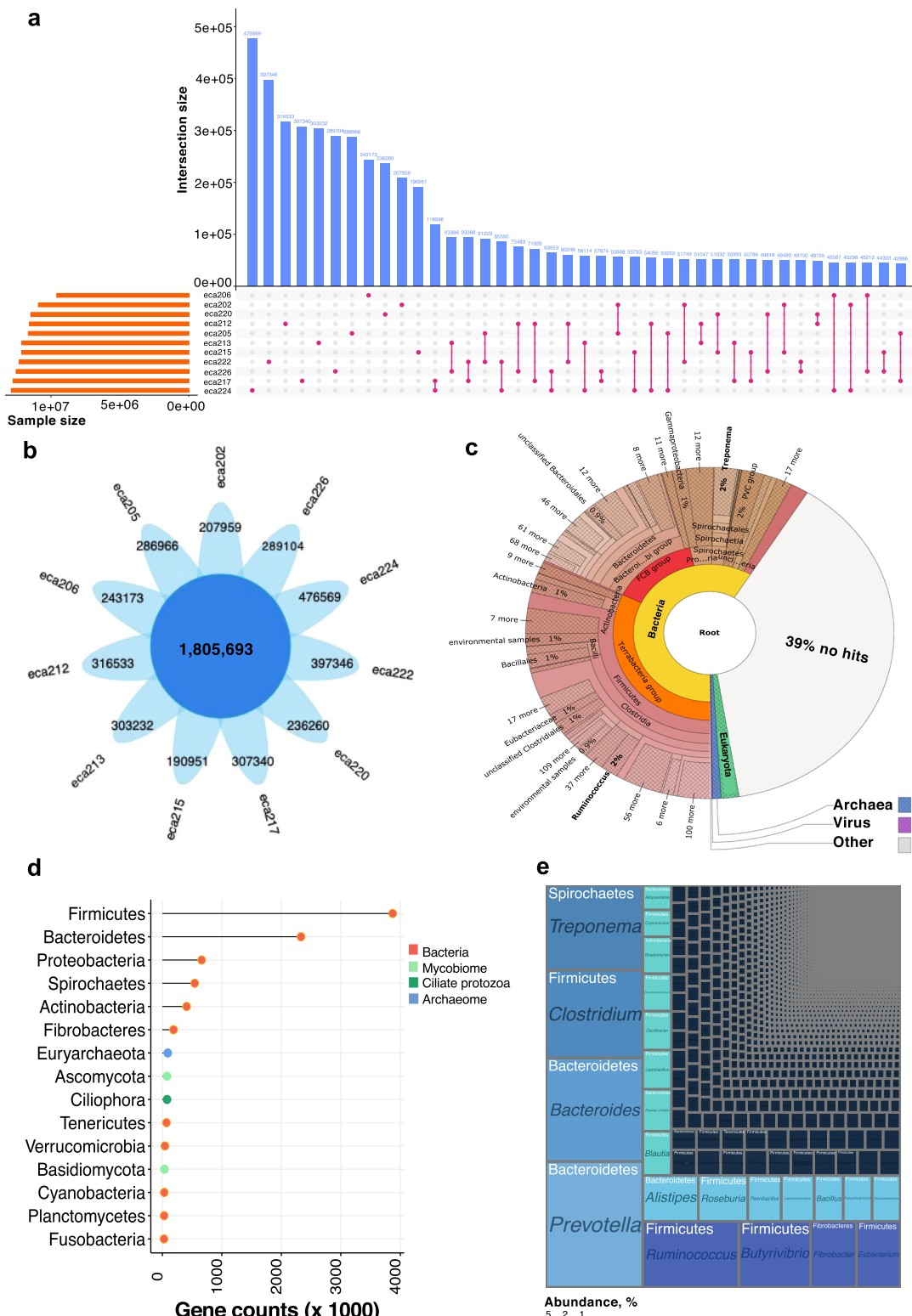

**Fig. 1 Taxonomic description of an expanded catalog of gut microbial genes: core genes and taxonomic annotation. a** Contribution of different sample sources (*n* = 11) to gene content of the horse gut microbial gene catalog. Vertical blue bars represent the number of genes present in only one sample or shared between pairs of samples. Horizontal orange bars in the lower panel indicate the total number of genes contained in each sample. **b** Flower plot showing the number of core genes shared between all samples (*n* = 11) and those specific for each individual. **c** Visualization of the taxonomic assignment of Illumina trimmed paired reads in a Krona plot using the software tool Kaiju. **d** Lollipop plot showing the gene counts identified by the Kaiju resolved at the phylum level. Dots are colored by kingdom. **e** Treemap showing the taxonomic ranking of the main taxa in the gene catalog using Kaiju. The size of each box is proportional to the number of genes assigned to each taxon. The placement of boxes is arbitrary concerning boxes within the same taxonomic rank and does not correspond to any form of phylogeny or relatedness.

phylum ($n = $ ~80,000 genes), stressing the importance and central role protozoa likely play in host function and microbiome metabolism (Fig. 1d). The most observed protozoa were *Stentor*, *Stylonychia*, *Paramecium*, *Ichthyophthirius*, and *Tetrahymena*. The taxonomic analysis also revealed that the core group of genes was defined by *Prevotella* (5.10%) and *Bacteroides* (2.98%), both dominant in the gut of marathon and triathlon athletes, respectively[33], along with *Ruminococcus* (3.17% of genes), *Treponema* (3.11%), *Clostridium* (2.70%), *Butyrivibrio* (2.42%), and *Fibrobacter* (1.37%; Fig. 1e; Supplementary Data 4). All of these fully agree with the core microbiota of horses inferred from 16S rRNA-based sequencing[24,34–36].

To gain functional insights, we annotated genes via KEGG orthologous groups (KOs) and carbohydrate-active enzymes (CAZymes) using the non-supervised orthologous groups (EggNOG) database. Results revealed a total of 12,060 KOs and 137 unique CAZymes, which encompassed 44% ($n = $ 11,132,404) and 3.38% ($n = $ 665,235) of the gene catalog, respectively. Most KOs had essential microbial gut functions, including metabolism (transport and metabolism of amino acids, carbohydrates, nucleotides, and lipids), genetic information processing, and signaling (Supplementary Data 5). We also identified potential functions related to the biosynthesis of secondary metabolites, quorum sensing, and prokaryotic defense system, which help maintain the microbiome's structure and the host's health (Fig. 2a). Given the abundance and variety of carbohydrates in the horse diet, the functionalities of glycosidic bond processing enzymes were investigated using the CAZy database. The great majority of these CAZYmes pertained to glycosyl-transferase (GT, 26.8%, $n = $ 228,878 genes) and glycoside hydrolase (GH, 64.9%, $n = $ 554,661 genes) families, followed by carbohydrate-binding modules (CBM; ~4%, $n = $ 37,474 genes) and polysaccharide lyases (PL) families (~1.6%, $n = $ 13,801; Supplementary Data 6). Confirming what has been recently found in horses[30], the most highly expressed CAZy family genes belonged to the GT2, GH5, and GH9 families, which contain diverse cellulases and hemicelluloses. Genes encoding GH13, GH57, GH77, and CBM48 families, characterized by their starch catalytic activities, were also highly represented in the catalog (Fig. 2b). Among 568,462 GHs and PLs genes, 53,569 (6.38%) were involved selectively in host glycan degradation (PL8, PL12, GH18, GH20, and GH38[37]; Supplementary Data 6).

We subsequently investigated the presence of antimicrobial resistance (AMR) genes within our gene catalog. A total of 57 clusters of AMR genes representing the major antibiotic classes were observed, including tetracycline ($n = $ 20), aminoglycosides ($n = $ 14), and macrolides, lincosamides, and streptogramins (MLS, $n = $ 9; Fig. 2c; Supplementary Data 7). The strong representation of Firmicutes and Bacteroidetes-associated tetracyclines resistance genes (*tet*(W), *tet*(Q), *tet*(O), *tet*(40)) and MLS resistance genes (*lnu*(C) and *mef*(A)) mirrored past observations in horses[30] and matched that found in humans[38], and livestock species such as cattle, pig, and chicken[39–41]. We detected an extended-spectrum of β-lactamase (ESBL) $bla_{ACI-1}$ found in several Negativicutes (Gram-negative Firmicutes) but rarely detected in animal or human gut microbiomes[42].

Last, we built 372 non-redundant prokaryotic MAGs at >50% completeness and contamination ≤10% (Supplementary Note, Supplementary Data 8, Supplementary Fig. 2a–h). Among these, 121 MAGs were estimated to be near complete; MAGs in this subset had minimal contamination (≤ 5%) and high completeness (> 95%). According to the Genome Taxonomy Database Toolkit (GTDB-Tk)[43], this MAG repertoire was assigned to 361 bacteria and 11 archaea, involving bacteria from the Bacteroidetes and Firmicutes phyla, followed by Spirochaetes, Euryarchaeota, Verrucomicrobia, Fibrobacteres, and Cyanobacteria phylum

(Fig. 2d). However, only 83 and 41 MAGs were classified at the genus and species levels, respectively (Fig. 2e). The MAGs pertaining to the Cyanobacteria, Proteobacteria, and Verrucomicrobia phyla displayed variable abundances between hosts (Fig. 2f). This trend was bolstered at the lower taxonomic level, except for MAGs assigned to *Fibrobacter* spp. (Supplementary Fig. 2h). Most MAGs encoded enzymes that degrade polysaccharides (Supplementary Note, Supplementary Data 8, Supplementary Fig. 3).

Notably, most MAGs ($n = $ 369) were new for horses[44,45], increasing the mappability of metagenomes and expanding our understanding of the horse microbiomes.

**The dominant basal gut phylotypes defined two taxonomical and functional microbial communities**. We first profiled microbial taxa in each metagenomic baseline sample at the taxonomic (NCBI *nr_euk* reference database) and functional (EggNOG database) levels. Then, we retained the most dominant microbial phylotypes. These phylotypes harbored 1,146 unique genera (accounting for 95% of the classified sequences) with abundance and prevalence in the top 25% and 50% quantiles, respectively (Supplementary Data 9). They were represented mainly by prokaryotes (91%), a handful of eukaryotes (7.15%), and archaea (1.75%). In agreement with 16S rRNA gene sequencing data from the same samples, *Prevotella, Fibrobacter, Clostridium Ruminococcus*, and *Treponema* were the most prevalent genera in most individuals (Supplementary Data 10 and 11). However, all individuals harbored large amounts of uncharacterized taxa in the horse, such as *Mediterraneibacter, Coprobacillus, Mucilaginibacter, Chitinophaga, Flavobacterium*, and *Enterocloster* (Fig. 3a).

The ordination analysis of these dominant microbial phylotypes yielded two distinct clusters of samples that recapitulated variation along the first axis (non-metric multidimensional scaling (NMDS), Bray–Curtis distances; Fig. 3b). The same pattern was supported by a permutational analysis of variance (pairwise PerMANOVA; $p = $ 0.008, $R^2 = $ 0.3716). Cluster 1 individuals ($n = $ 3 horses) exhibited higher α-diversity despite the small sample size (Shannon and inverse Simpson indices; $p = $ 0.0134 for both, two-sided Wilcoxon rank-sum test, Fig. 3c, d). We then investigated which microbial taxa underpinned each cluster. Cluster 1 mainly included the enrichment of many poorly-characterized taxa ($n = $ 318; IQR: $2.90e^{-04}$– $1.16e^{-04}$; Fig. 3e, f) that encompassed Proteobacteria ($n = $ 132), Actinobacteria ($n = $ 49), Verrucomicrobia ($n = $ 17), Planctomycetes ($n = $ 17), and Cyanobacteria ($n = $ 7). Methanogens previously described in the bovine rumen[46,47]—including those belonging to the Methanomicrobiales (*Methanothrix*), Methanobacteriales (*Methanosphaera, Methanobacterium, Methanothermobacter*), and Thermococcales (*Thermococcus*) orders—were also more abundant in that cluster (DESeq2, adjusted $p < $ 0.05; Supplementary Data 12). In contrast to the overwhelming diversity observed in cluster 1, a dwarfed biodiversity defined cluster 2 that assembled around highly efficient fiber degraders (members from the *Lachnospiraceae* taxa: *Anaerostipes, Butyrivibrio, Blautia, Coprococcus, Dorea, Eubacterium, Hespellia Lachnospira, Oribacterium, Roseburia*, and L-*Ruminococcus*) and downstream users of degradation products (*Prevotella*, and *Treponema*)[47] (DESeq2, adjusted $p < $ 0.05; Supplementary Data 12, Supplementary Fig. 4a–d).

The sample distribution based on microbial KOs (PerMANOVA; $R^2 = $ 0.3725, $p = $ 0.008) and CAZymes (PerMANOVA; $R^2 = $ 0.6063, $p = $ 0.005) echoed that of taxonomical composition, supporting taxa segregation by their underlying biochemical activities (Supplementary Fig. 5). Cluster 1 captured higher

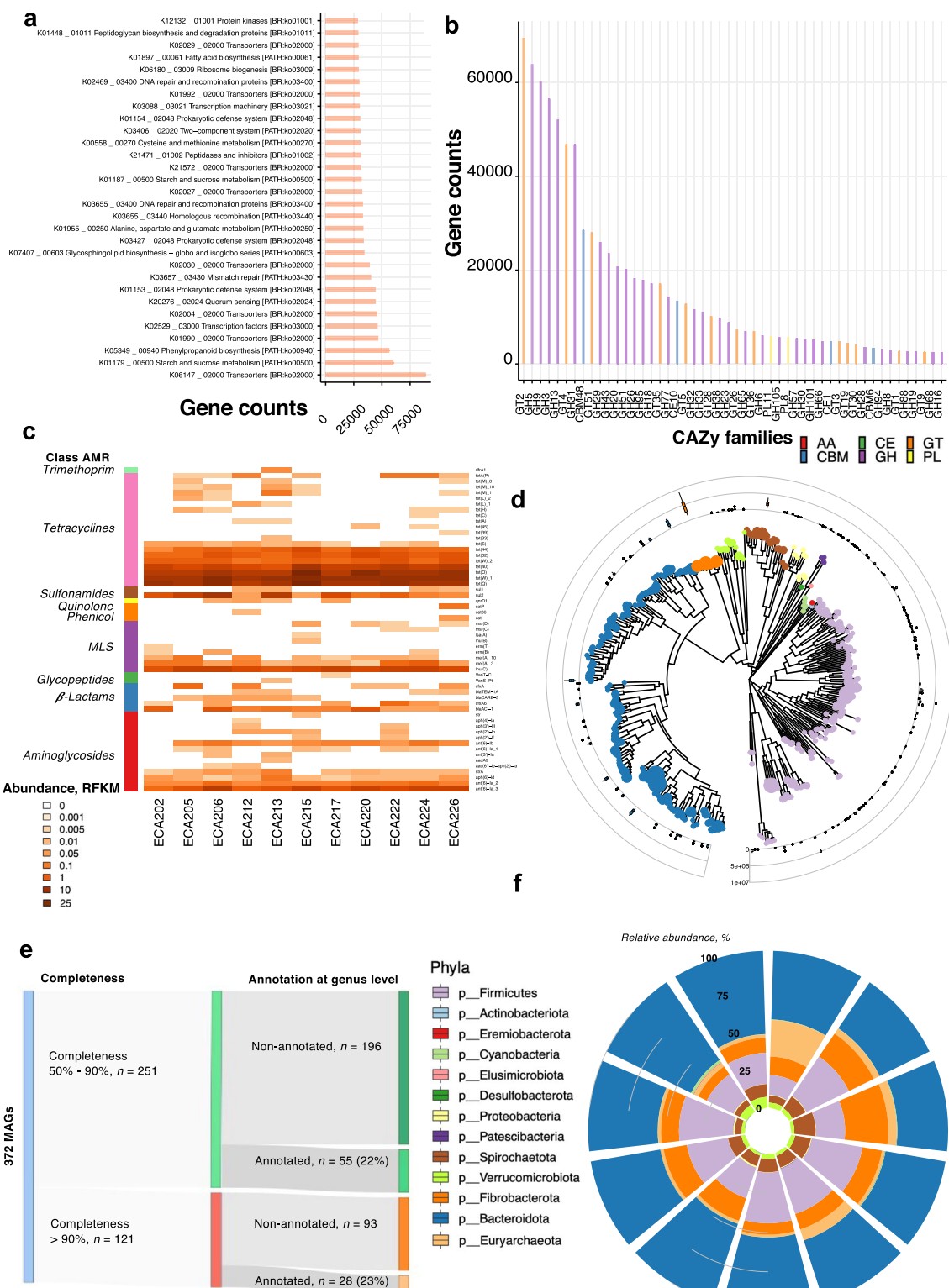

**Fig. 2 Functional diversity of the equine microbial genes catalog, antimicrobial genes, and MAGs landscape. a** Frequency of KEGGs pathways in the gut microbial gene catalog estimated from the orthologous groups (KOs). The horizontal bars represent the absolute number of genes found in each KEGG pathway. **b** The Frequency of total carbohydrate-active enzyme (CAZymes) families estimated in the gut microbial gene catalog. CAZymes are colored according to their class. **c** Heatmap shows each individual's (n = 11) normalized counts of antimicrobial resistance (AMR) genes based on the ResFinder database. The left column depicts the AMR class. **d** The Phylogenetic tree of the n = 372 MAGs detected. The outer cycle boxplots represent the abundance of each MAG. Boxplots are colored according to phyla. **e** Sankey diagram showing the numbers of MAGS with completeness between 50 and 90% or > 90%. Only a relatively small fraction of the MAGs (73 out of 372) was annotated at the genus level. **f** Circular stacked bar plot of the MAGs phyla abundance detected for each individual in the cohort (n = 11).

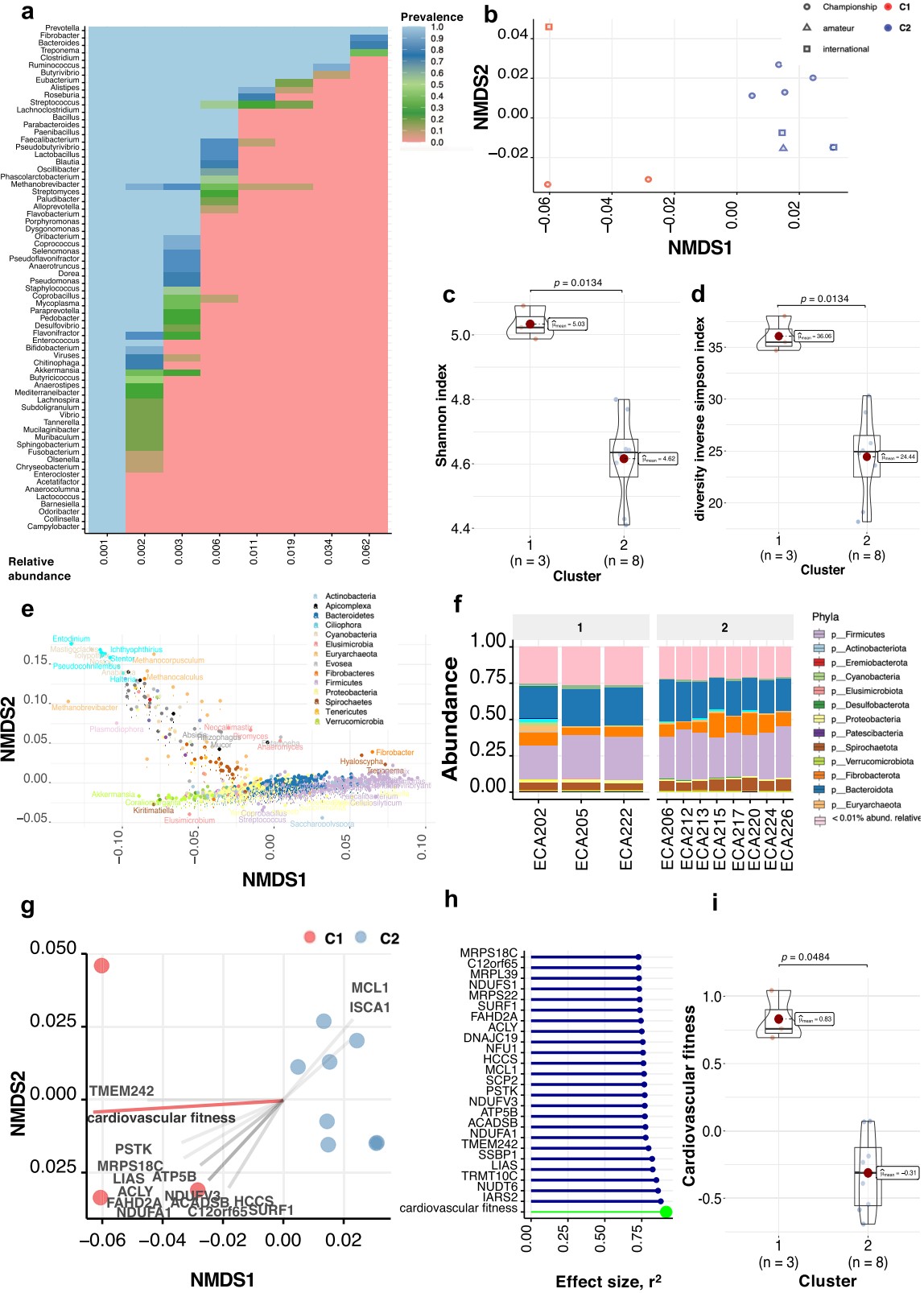

CAZyme diversity. It showed enrichment of CAZymes that might act in a concerted manner to cleave plant polysaccharides into fermentable monosaccharides (GH6, GH39, GH54, GH82, GH84, GH99) or host glycosaminoglycans (PL8, PL12, GH18, GH20, GH29, GH88, and GH95; DESeq2, adjusted $p < 0.05$; Supplementary Data 13, Supplementary Fig. 4d). Cluster 1 also had

more CAZymes for host glycans degradation (14.15%) than members of cluster 2 (9.63%). In agreement, the latter presented an enrichment of KOs related to glycan biosynthesis and metabolism (K09953, K18770, K12309, K12551, K01137, K03276, K12985, K14459). Cluster 1 also exhibited large metabolic increases ($n = 520$ KOs) in amino acids metabolism

**Fig. 3 Basal dominant gut phylotypes composition is associated with cardiovascular fitness. a** Dominant phylome and its prevalence at different detection thresholds (relative abundance). The percentage of shared items and the proportion of shared samples are represented on the y- and x-axis. **b** NMDS ordination analysis (Bray–Curtis distance) of dominant phylotype composition. Points denote individual samples ($n = 11$), colored according to the clustering group. The shape of the dots indicates the competition level of horses. **c, d** Violin plot representing Shannon Diversity Index and inverse Simpson index of dominant phylotypes according to the clustering groups, respectively. In all cases, colors indicate community classification, cluster 1 (red color, $n = 3$) and cluster 2 (blue color, $n = 8$). Boxplots show the median, 25th, and 75th percentile, the whiskers indicate the minima and maxima, and the points lying outside the whiskers of boxplots represent the outliers. Adjusted p-values from two-sided Wilcoxon rank-sum test. **e** Biplot values of the discriminant phylotypes driving the NMDS ordination. The phylotypes contributing to the distinction between groups on at least one axis are depicted. Points are colored by phylum. **f** Taxonomic distribution of the dominant phylotypes grouped by phyla in each individual ($n = 11$). Individuals are split by cluster: cluster 1 ($n = 3$) and cluster 2 ($n = 8$). **g** NMDS ordination plot shows the covariates contributing significantly to the variation of dominant phylotypes determined by the envfit() function. The arrows for each variable indicate the direction of the effect and are scaled by the unconditioned $r^2$ value. Dots represent samples ($n = 11$), which are colored according to the type of community: cluster 1 (red color, $n = 3$) and cluster 2 (blue color, $n = 8$). **h** Effect sizes of the main variables affecting the NMDS ordination. The length of the horizontal bars shows the amount of variance ($r^2$) explained by each covariate in the model. Covariates are colored according to the type of dataset: athletic performance are in green and mitochondrial-related genes in blue. **i** Violin plot representing the cardiovascular fitness of each cluster, which was calculated as a composite of post-exercise heart rate, cardiac recovery time, and average speed during the race. Colors indicate community classification, cluster 1 (red color, $n = 3$) and cluster 2 (blue color, $n = 8$). The boxplots show the median, 25th, and 75th percentile, and the whiskers indicate the minima and maxima. The points lying outside the whiskers of boxplots represent the outliers. Adjusted p-values from the two-sided Wilcoxon rank-sum test.

(K12256, K15226, K03852, K08688, K18049, K21062), energy production, e.g., genes within lipid metabolism (K10781, K16795, K01795, K01049, K12309), propanoate metabolism (K11264, K01659 K00382 K01720), fructose and mannose (K19633) and pyruvate (K02594), as well as methane production (K00202, K00204, K00583, K05884; DESeq2, adjusted $p < 0.05$; $\log_2$FC|>1.5|, Supplementary Data 14). In accordance with protein catabolism, plasma *iso*-valerate—a precursor for muscle glycogen[1]—was also higher in cluster 1 individuals ($p = 0.0121$, two-sided Wilcoxon rank-sum test). These findings strongly suggest that cluster 1 horses had a larger substrate range. However, this highly functional complexity could not be captured at the MAG level (Supplementary Note, Supplementary Data 8, Supplementary Fig. 6a–d).

Cluster 1 metagenomes also had reduced representation of the host DNA in the raw sequencing data ($p = 0.05$, two-sided Wilcoxon rank-sum test, Supplementary Fig. 5, Supplementary Data 2), suggesting reduced epithelium disruption or intestinal inflammation, commonly observed in endurance athletes[48].

**Basal dominant gut phylotypes were closely associated with cardiovascular fitness in equine athletes.** To tie together the basal dominant gut phylotypes with the horse performances, we investigated whether diversity, compositional and functional differences between clusters were driven by nutrient intake or any horse-centered parameter. Recorded dietary data showed that participants had a constant standard varied macronutrient diet on training and non-training days [mean ± SD (% dry matter (DM) basis): 53.7 ± 19.3 MJ/day, 939 ± 42 g/day of protein (11 ± 1.9%), 678 ± 20 g/day of fat (8 ± 1.6%), 2190 ± 706 g/day of hydrolyzable carbohydrates (26 ± 8.9%), and 2320 ± 877 g of fermentable fiber (27 ± 2.8%)]. Individuals were fed 8.43 ± 2.99 kg/day, with an 80:20 forage to concentrate ratio (Supplementary Data 1). None of the macronutrient intakes were statistically different between the two clusters ($p > 0.05$, two-sided Wilcoxon rank-sum test), and none of them were the major driving forces of the microbiome composition (PerMANOVA; $R^2 = 0.102 ± 0.089$ and p-values ranging from 0.055 to 0.82; Supplementary Fig. 7a–d). Likewise, KOs (PerMANOVA; $R^2 = 0.034 ± 0.010$ and p-values ranging from 0.940 to 0.969) and CAZymes profiles (PerMANOVA; $R^2 = 0.030 ± 0.016$ and p-values ranging from 0.250 to 0.971) were also independent of nutrient intakes. Gut community composition was neither linked to horse sex (PerMANOVA; $R^2 = 0.2486$, $p = 0.266$), or age of the athletes (PerMANOVA; $R^2 = 0.0952$, $p = 0.409$; Supplementary Fig. 7e–g). Moreover,

clusters 1 and 2 did not overlap with the horse kinship (Supplementary Fig. 8).

After confirming that these confounding factors did not define the gut microbiome phylotypes, we next tested if any host-centered omic or phenomics dataset, including transcriptomics, metabolomics, blood biochemical assay profiles, acylcarnitines, and cardiovascular fitness parameters, best captured the distributions of metagenomic samples (Supplementary Note, Supplementary Data 15–18, respectively). Cardiovascular fitness—a composite variable of post-exercise heart rate, cardiac recovery time, and average speed during the race—was the primary driver of the overall structural variation of the gut metagenome (*envfit*, $R^2 = 0.9192$, adjusted $p = 0.005$; Supplementary Data 19).

This cardiovascular composite parameter aggregated 39.64% of fecal microbiome community variation, thereby outperforming the expression of several mitochondrial-related genes (Fig. 3g, h). Horses from cluster 1 had significantly higher cardiovascular fitness than cluster 2 members ($p = 0.048$, two-sided Wilcoxon rank-sum test, Fig. 3i). That did not increase blood lactate concentration ($p = 0.921$, two-sided Wilcoxon rank-sum test), which is a proxy for glycolytic stress and disturbance in cellular homeostasis[1].

**An independent validation set confirmed that *Lachnospiraceae* bacteria were negatively associated with cardiovascular fitness in highly trained equine athletes.** We further attempted to validate this association with 16S rRNA sequence data from the gut microbiota of 22 independent highly trained endurance horses (Supplementary Data 20, 21)

As with the study cohort, the microorganisms' community profiles could be distinguished based on the horse's cardiovascular fitness (pairwise PerMANOVA on a Bray–Curtis distance matrix; L vs. H: adjusted $p = 0.057$, $R^2 = 0.109$, Supplementary Fig. 9a, b), with higher microbiota dispersion in more fit athletes (betadisper(), L vs. H: adjusted $p = 0.0243$; Supplementary Fig. 9c). We used sparse Partial Least Squares-Discriminant Analysis (sPLS-DA) to find a taxa panel that discriminated between the more and less fit horses (Supplementary Fig. 9d). This analysis only supported the presence of *Mogibacterium* and the yet undefined members of *Rhodospirillaceae, Enterobacteriaceae, Planococcaceae,* and *Sphingobacteriaceae* families in more fit individuals but not the other 252 poorly-described bacteria members. This likely reflects an under-representation of these taxa in existing 16S rRNA reference databases. Conversely, the sPLS-DA corroborated the enrichment of Clostridiales,

*Erysipelotrichaceae*, and *Lachnospiraceae* taxa in less competitive individuals. Markedly, genera such as *Blautia, Butyrivibrio, Coprococcus, Dorea, Desulfovibrio, Hespellia, Lachnospira*, and *L-Ruminococcus* (all pertaining to the *Lachnospiraceae* family) were consistently depleted in less fit athletes in both the discovery and the validation sets. In addition, *Dorea* and the polyphyletic *Ruminococcus* were consistent hallmark genera of less competitive individuals (DESeq2, adjusted $p = 0.0491$; Supplementary Fig. 9e, f) across both discovery and validation cohorts. This further strengthens the association between the horse's cardiovascular fitness and gut microbiota composition.

Although larger metagenomic cohorts and improved reference collections are required to validate the relationship between athletic performance and gut microbiome, these data do, however, confirm the negative association between Firmicutes (notably *Lachnospiraceae* taxa) and cardiovascular fitness in endurance athletes.

**Holo-omics: microbiomes with a higher prevalence of *Lachnospiraceae* taxa signed lower cardiovascular fitness and pointed toward likely impaired mitochondrial capacity.** To further characterize the microbiome-host crosstalk and identify molecular differences between the two types of cardiovascular outcomes in elite horses, we integrated multi-omic datasets from the host and associated gut microorganisms through a multivariate matrix factorization approach (DIABLO; see "Methods"). To achieve this integrated perspective coined as holo-omics[49], we combined host-centered omic and phenomics data with the fecal shotgun metagenomics, SCFAs composition, and the concentrations of bacteria, anaerobic fungi, and protozoa.

First, we observed strong covariation between the dominant phylotypes and the genetic functionalities derived from KOs ($r^2 = 0.99$) and CAZymes ($r^2 = 0.98$; Fig. 4a). This apparent correlation supports the added value of microbiome functions for status prediction rather than composition alone, as noted in human athletes[21,50]. Concomitantly, the microbiome composition highly covaried with the mitochondrial transcriptome ($r^2 > 0.80$) and loads of fecal protozoa ($r^2 > 0.80$; Fig. 4a). The targeted (acylcarnitine profiles and biochemical assays) and untargeted host metabolomic analysis (1H NMR) also accounted for the metagenome variation, albeit of lesser significance ($r^2 = 0.57-0.61$; Fig. 4a). Then, to add biological meaning to the predicted model, we investigated the relationship between the DIABLO-selected features with the highest covariation (Supplementary Fig. 10a–c). The first latent variable of the predicted model indicated that athletes with higher cardiovascular fitness harbored a wide range of multi-kingdom and yet undescribed clades in horses (Fig. 4b). Consistent with the univariate analysis, it included the facultative bacterial predator *Lysobacter, Akkermansia*, generally regarded as health-promoting bacteria in athletes[51–55], along with anaerobic fungi (*Ophiocordyceps, Pseudogymnoascus, Trichoderma, Talaromyces, Rhodotorula, Exophiala, Puccinia*), methanogens (*Methanothermobacter, Methanothrix*) and algae (*Emiliania* and *Porphyra*). It is worth noting that the ciliate protozoa, at up to 18% of the biomass (~$10^9$ cells/g of stool), were discriminative of more fit individuals (Supplementary Fig. 10d). Conversely, less fit subjects were largely defined by members of the family *Lachnospiraceae, Treponema, Prevotella*, and other commonly described taxa in horses, including Clostridiales and undescribed *Erysipelotrichaceae* taxa. Paired with this less complex community, the first latent variable pointed at a redundant functional diversity, spanning CAZymes to extract energy from recalcitrant polysaccharides (GH8, GT36, GH51, GH28, GT2, GH5, GH3; Fig. 4c). While intestinal microbiota members belonging to the

*Lachnospiraceae* family are known to produce acetate and butyrate[56], none of these SCFA were significantly increased in the feces or plasma of these athletes ($p > 0.05$; two-sided Wilcoxon rank-sum test), and the fecal pH remained unchanged ($p = 0.837$; two-sided Wilcoxon rank-sum test; Supplementary Data 22).

At the transcriptome level, the first latent variable supported an impairment rather than improved metabolic flexibility in the less fit individuals. The latter exhibited downregulation of mitochondrial-related genes involved in β-oxidation (*ECI1, SCP2, ACLY*), electron transport chain (*TMEM242, NDFB4, TMEM126B, NDUFV3, NDUFA1, NDUFA10, SURF1, NDUFV1, DLD*), Ca²⁺ translocation (*PMPCA, VDAC2, PHB*), mitophagy (*TOMM40*), and biogenesis (*SSBP1, ACSS2*; Supplementary Fig. 10e). These results suggested decreased mitochondrial fatty acid oxidation and increased glucose catabolism, which progressively impedes longer running times, and fatigue resistance. In agreement with this notion, less fit individuals presented reduced concentrations of glucose ($p = 0.0484$, two-sided Wilcoxon rank-sum test), increased accumulation of long-chain acylcarnitines in plasma (e.g., oleoyl carnitine, $p = 0.0484$; hydroxy oleoyl carnitine, $p = 0.0242$, two-sided Wilcoxon rank-sum test) and a tendency for augmented non-esterified fatty acids (NEFAs) in plasma ($p = 0.0848$, two-sided Wilcoxon rank-sum test). Therefore, less diverse microbial communities dominated by a few Firmicutes-derived families (mainly *Lachnospiraceae*) could constrain the horse mitochondrial aerobic ATP production for extended cardiovascular fitness.

**Frenemies: *Lachnospiraceae* and poorly-described phylotypes in horse athletes.** To gain insight into the co-occurrence and co-exclusion relationships between multi-kingdom microbial genera and functions, we applied an inverse covariance estimation for ecological associations. This approach identified 12 modules. Among them, we uncovered two extreme assortative modules characterized by robust microorganism-microorganism or microorganism-functions interactions (Fig. 4d) that agreed with the other univariate and multivariate analytical frameworks. The first module was mainly characterized by bacterial interactions within the Firmicutes phylum (mainly from the *Lachnospiraceae* family) and CAZy families that can target the substrate of plant structural polysaccharides (GH3, GH39, GH51, GH82, GH84). On the other hand, the second extreme network encompassed widespread yet undescribed bacteria in horses from the Proteobacteria, Actinobacteria, Planctomycetes, and Verrucomicrobia, together with CAZymes active in the degradation of the complex structure of plant cell-wall materials (GH28) and host glycans (GH20, GH18, GH33; Fig. 4d).

## Discussion

The current study presents a comprehensive horse gut microbiome gene catalog and its association with endurance performance. The more than 25 million non-redundant genes in this catalog have widened 20-fold the number of genera known to reside in the gastrointestinal tract of horses[20,32,44,45,57,58], uncovering a substantial number of bacteria ($n = 2927$), archaea ($n = 174$), and Eukaryota ($n = 1081$). This catalog also captured a wide array of functions related to complex carbohydrate fermentation and functional capacities that enabled more energy extraction from dietary, microbial, and host resources, which likely provided a performance advantage for athletes. For example, genes encoding enzymes for glycosidic-bond cleavage (GHs and PLs) represented the majority (67%) of the CAZyme genes, highlighting the indispensable role of gut microorganisms in glycan metabolism. Intriguingly, 6.38% of GHs and PLs were

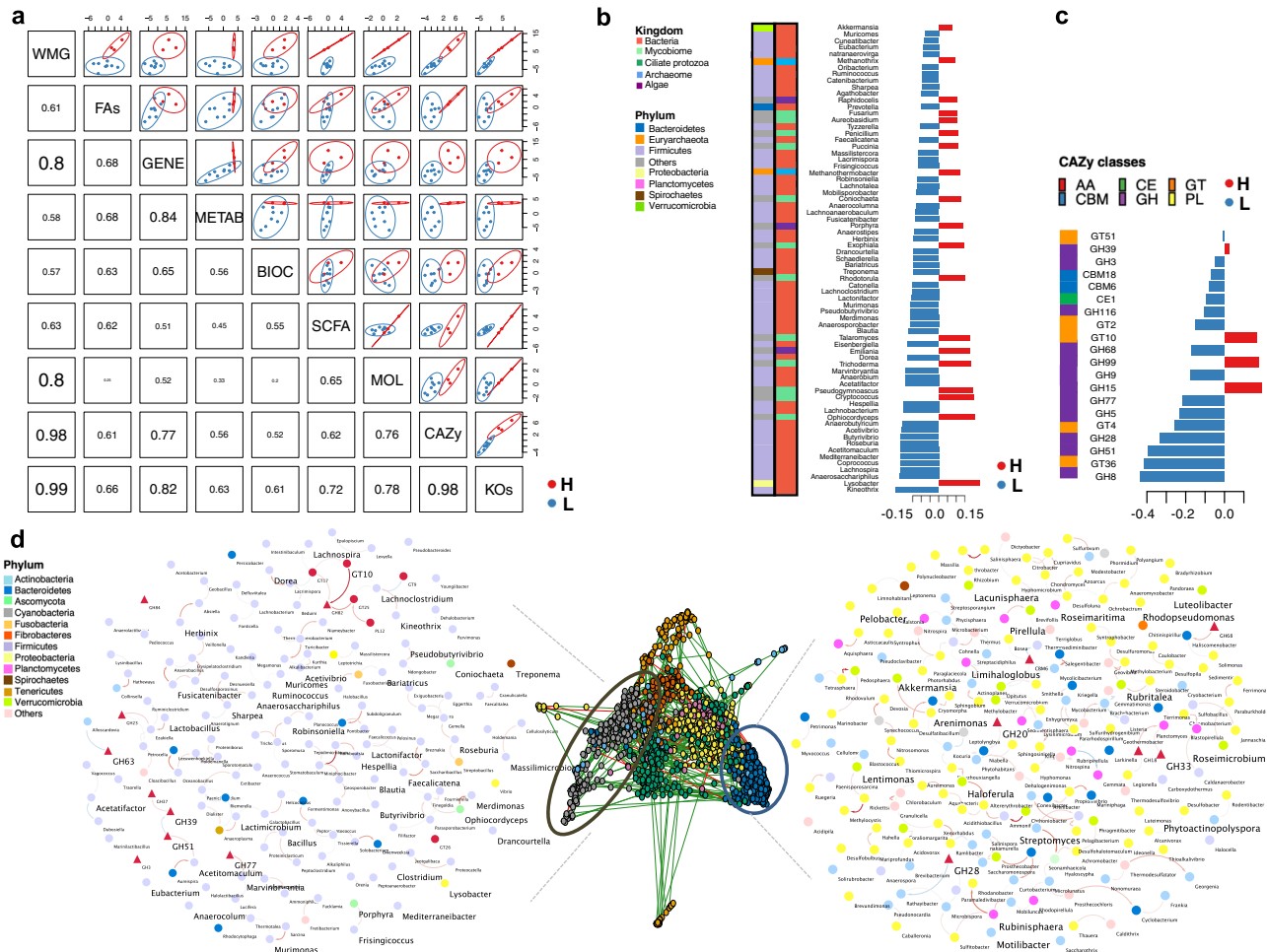

**Fig. 4 Sportomics: data integration supports the link between cardiovascular fitness and the gut microbiome composition and functionality. a** Matrix scatterplot showing the correlation between the first components related to each dataset in DIABLO according to the input design. **b** Dominant microbial genera contributing to the separation along with component 1 of the microbiome dataset. Microbiome data are centered log-ratio-transformed, and bar length indicates loading coefficient weight of selected phylotypes, ranked by importance, bottom to top. Columns on the left depict the kingdom and phylum of each discriminant phylotype. **c** CAZymes contributing to separation along with component 1 of the CAZy dataset. CAZymes profiles are log-transformed median-scaled values. Bar length indicates the loading coefficient weight of selected CAZymes, ranked by importance, bottom to top. In all cases, colors indicate community classification, cluster 1 (red color, n = 3) and cluster 2 (blue color, n = 8). The column on the left depicts the CAZymes class. **d** Co-occurrence network analysis of dominant phylotypes and carbohydrate-active enzymes (CAZy) types datasets using sparse inverse covariance estimation for ecological association inference (SPIEC-EASI). Louvain clustering was able to generate 12 feature co-occurrence modules. The two extreme assortative modules are depicted in detail using Cytoscape. A positive correlation between nodes is indicated by red connecting lines, and a negative correlation by blue. A circle or triangle denotes microbial clades and CAZymes features. Microbial nodes are colored by phyla. Elements with larger text sizes are those revealed as discriminant along with component 1 by the DIABLO approach. Edge width corresponds to the strength of the association between features.

involved selectively in cleaving endogenous host-related glycosylated proteins and glycans[37]. It is yet unclear whether these CAZymes correspond simply to the peptidoglycan-processing machinery required for bacterial cell growth and division or whether peptidoglycan and highly glycosylated proteins represent a major carbon source for some bacteria or the host after being transformed to acetate[59].

In addition, we have identified a set of 372 MAGs. The number of microbial genes and MAGs will likely increase as more samples are analyzed, as seen in equids[44,45] and other livestock gut metagenomes[29,60–62]. Nonetheless, the present metagenomic genes and MAGs repertoire are a step forward in explaining the composition and function of the horse gut microbiome, especially in the context of endurance exercise.

Along with fatigue resistance, cardiovascular fitness is a crucial indicator of endurance performance in human athletes[9]. In the

considered elite endurance horses, the cardiovascular capacity defined about ~40% of the variation in the most dominant gut phylotypes (e.g., most abundant and ubiquitous) in the absence of any confounders. This association was replicated in an independent cohort of elite horses and outweighed that estimated between marathon runners' cardiovascular fitness and their microbiota (~22% of explained variance[22]). Moreover, we found that higher cardiovascular capacity was mostly associated with uncharacterized microbial features and functional pathways. Cardiorespiratory fitness matched increased gut microbiota diversity in marathon runners[21]. A possible explanation for this phenomenon could stem from the broader use of resource compounds, including SCFAs, host glycans, dietary and microbial proteins, complex lipids, propanoate, pyruvate, and hydrogen, to produce higher relevant goods for host energy requirements. This notion was exemplified by the extensive enrichment of CAZymes

and KOs involved in the metabolism of glycans, amino acids, carbohydrates, and lipids.

The deep phenomics applied to these horses pointed to the microbiome-mitochondria crosstalk as a potentially highly effective way to modulate cardiovascular capacity. Conceptually, the higher availability of goods likely influenced the mitochondria function and readouts. This hypothesis paralleled increased glucose availability in serum, a precursor for glycogen in muscle, alongside reduced circulating acylcarnitines and NEFA levels. The acylcarnitine and NEFA drop was possibly not due to reduced substrate availability but to an increase in transport into the mitochondria through the carnitine shuttle, typically for ß-oxidation. In concert, mitochondrial-related genes involved in energy resilience, biogenesis, and $Ca^{2+}$ cytosolic transport were simultaneously upregulated in participants with greater cardiovascular capacity suggesting enhanced mitochondrial oxidative phosphorylation and FAO, the two metabolic pathways central to energy production[63]. Therefore, bidirectional regulatory circuits between host and inter-microbial components can seemingly increase mitochondrial substrate availability to meet the high energy needs during exertion while optimizing cardiovascular capacity, longer running times, and fatigue resistance. Beyond its energy-producing ability, mitochondria are essential for the physiological activity of the cardiovascular system due to their crucial role in the regulation of intracellular $Ca^{2+}$ fluxes, which contribute to cardiac muscle contraction[64].

Methane-producing taxa were over-represented in more fit individuals' gut microbiomes. Methane cannot be absorbed by animals[65], but might exert anti-inflammatory, anti-apoptotic, and anti-oxidative effects in the gut[66]. This means that methane-producing taxa that primarily contribute to energy waste may counterintuitively help to confer gut protection from the inflammatory effects of endurance exercise[67]. Supporting this notion, the best performers showed reduced amounts of horse DNA in the feces, a proxy of epithelial cells and leukocytes' shedding. Whether methane production is a host-microbiome engineering adaptation to reduce intestinal permeability, disruption, and inflammation remains an open question.

On the other hand, the microbiomes of less fit individuals exhibited lower diversity. They were dominated by specialized cellulose and hemicellulose degraders of the *Lachnospiraceae* family. Despite this specialization, SCFA concentration was similar to the more fit individuals. A similar observation was made in humans with low aerobic fitness whose microbiota harbored more *Eubacterium rectale-Clostridium*[23]. Therefore, in light of these findings, nutritional interventions to reduce *Lachnospiraceae* taxa like *Dorea* and L-*Ruminococcus* while creating more space for non-core species will likely be required to increase microbiome diversity, functional plasticity, and athletic performance. Dietary changes can promote swift changes in *Lachnospiraceae* abundance[57]. Consumption of dietary pectins could strongly reduce the fecal abundance of *Ruminococcus*, *Blautia*, and *Roseburia* (all pertaining to the *Lachnospiraceae family*)[68]. Conceptually, the partial replacement of diet-derived polysaccharides needed to expand *Lachnospiraceae* taxa (e.g., starch, inulin, xylans, and arabinoxylan) by other structurally diverse dietary fibers (e.g., soluble pectins such galactan, arabinan, and arabinogalactan) could be the front-line for nutritional interventions. Due to the minor involvement of the *Lachnospiraceae* phylotypes in proteolytic metabolism[69], increasing protein consumption could be another way to optimize the microbiome function in horse athletes.

The present study contains several limitations. First, interactions should be interpreted cautiously, and the associations cannot be considered direct causal effects. Second, due to the difficulty of sampling elite endurance horses after long races, our observations were made from a small sample size on a single race, with a few samples for independent validation. Replicating results in other athletes with continuous long-term measurements will hence be needed to assess the generalizability of these findings. Third, the current study did not include negative control samples to control for DNA contamination before or during shotgun sequencing. Still, our analyses were restricted to the most prevalent and dominant phylotypes to limit inaccuracy in further data interpretation. Last, functional investigations to identify microbial and host metabolites through $^{1}H$ NMR and mass spectrometry coupled with longitudinal meta-transcriptomics and metagenomics are needed to improve inference of microorganism's functionalities and support the herein reported findings.

Beyond these limitations, the present study presented several strengths. First, this study revealed an unprecedented level of microbial diversity, biotic interactions, and functional gene potential in the gut of horse athletes. Second, holobiont data integration suggested that the variability of the gut microbiome composition and functions was associated with cardiovascular fitness in two different ways. On the one hand, less diverse microbial communities comprising high amounts of *Lachnospiraceae* taxa showed high functional microbial redundancy and downregulation of mitochondrial-related genes associated with energy production, biogenesis, and $Ca^{2+}$ translocation, thereby leading to reduced amounts of aerobic ATP, impaired cardiovascular function, and thus reduced athletic performance. On the other hand, gut communities harboring an extensive range of yet undescribed phylotypes in horses, including a myriad of anaerobic fungi, methanogens, and protozoa, were metabolically more active and offered complimentary or unique metabolic pathways to enhance fuel bioavailability for the mitochondria while improving aerobic work competences, sparing glycogen usage, and increasing cardiovascular capacity.

The athletes with the greatest improved cardiovascular fitness likely have better fatigue resistance, a critical factor in achieving competitive success. Functional studies of gut microbiome species linked to mitochondria function will be instrumental in developing dietary strategies that optimize cardiovascular capacity and, therefore, athletic performance.

## Methods

**Ethical approval**. The local animal care approved the study protocol and use committee (ComEth EnvA-Upec-ANSES, reference: 11-0041, dated July 12, 2011), and protocols were conducted following the EU regulation (nº 2010/63/UE). Owners and riders provided their informed consent before the start of sampling procedures with the animals. The horses (*Equus caballus*) used in this research study were pure-breed or half-breed Arabian (three females, one male, and seven geldings; age: 10 ± 1.69 years old).

**Animals**. Eleven endurance horses were selected from a cohort previously used in our team[6,24,25,70]. All equine athletes started training for endurance competitions at age 4 and presented a similar training history, level of physical fitness, and training environment. The 11 horses were selected due to the following criteria: (1) enrollment in the same 160 km endurance category; (2) blood sample collection before and after the race; (3) feces collection before the race; (4) absence of gastrointestinal disorders during the four months before enrollment; (5) absence of antibiotic treatment during the four months before enrollment and absence of anthelmintic medication within 60 days before the race, and (6) a complete questionnaire about diet composition and intake.

Subject metadata, including morphometric characteristics and daily macronutrient diet intake records, is depicted in Supplementary Data 1. Daily nutrient intake calculations are described elsewhere[24].

**Performance measurement**. The endurance race was split into ~30–40 km phases. At the end of each phase, veterinarians checked horses (referred to as a vet gate). The heart recovery time was the primary criterion evaluated at the vet gate as it is shown to be an excellent complement to a physical assessment of an individual. The heart rate was measured at each vet gate by the riders and a veterinarian using a heart rate meter and a stethoscope. Any horse deemed unfit to continue (due to a

heart rate above 64 bpm after 20 min of recovery) was immediately withdrawn from the event.

It should be noted that the time interval between arrival at the vet gate and the time needed to decrease the heart rate below 64 bpm was counted as part of the overall riding time. Therefore, the cardiac recovery time was calculated as the difference between the arrival time (at the end of the phase) and the time of veterinary inspection (referred to as the "time in" by the FEI endurance rules). The average speed of each successive phase was calculated at the vet gate.

Changes in these three variables during endurance events have been shown to predict whether a horse is aerobically fit or not[71]. We consider these variables to estimate cardiovascular capacity linked to performance capability and achievement. Therefore, these three variables were first scaled through a Z-score; that is, the number of standard deviation units a horse's score is below or above the average score. Such a computation creates a unitless score that is no longer related to the original units of analysis (e.g., minutes, beats, Km/h). It measures the number of standard deviation units and can more readily be used for comparisons. A composite based on such Z-scores was then created to estimate cardiovascular fitness. Specifically, the composite() function of the multicon R package (v.1.6) was used to develop a unit-weighted composite of the three variables listed above.

**Estimation of the pedigree kinship matrix**. The kinship2[72] (v.1.8.5) R package was used to calculate the pedigree kinship matrix of all individual pairs, plot the pedigree, and trim the pedigree object. The kinship coefficient for any two subjects was calculated as the probability that an allele chosen at random for both subjects at a given locus is identical-by-descent, that is, inherited from a common ancestor[72]. The pedigree was calculated using six generations back for the 11 Arabian horses of the study. The pedigree kinship matrix was then visualized using the plot_popkin() function from the popkin (v.1.3.17) R package. The inbr_diag() function was used to modify the kinship matrix, with inbreeding coefficients along the diagonal, preserving column and row names.

**Blood sampling**. Blood samples were collected from each horse the day before the event (Basal, T0) and immediately after the end of the competition (T1) for transcriptomic, biochemical, metabolomic, and acylcarnitine assays. As described elsewhere[24], pretreatment of the blood samples was carried out immediately after the collection because field conditions provided access to refrigeration and electrical power supply. Briefly, blood samples for RNA extraction were collected using Tempus Blood RNA tubes (Thermo Fisher) and stored at −80 °C. Whole blood samples were taken in EDTA tubes (10 mL; Becton Dickinson, Franklin Lakes, NJ, USA) to determine biochemical parameters, while for the metabolome profiling, the sodium fluoride and oxalate tubes were used to inhibit further glycolysis that may increase lactate levels after sampling. Then, clotting time at 4 °C was strictly controlled for all samples to avoid cell lyses that affect metabolome components. After clotting at 4 °C, the plasma was separated from the blood cells, transported to the lab at 4 °C, and frozen at −80 °C (no more than 5 h later, in all cases). Concerning the acylcarnitine, blood samples were collected in plain tubes. After clotting, the tubes were centrifuged, and the harvested serum was stored at 4 °C for no more than 48 h and subsequently stored at −80 °C.

**Transcriptomic microarray data production, pre-preprocessing, and analysis**. According to the manufacturer's instructions, total RNAs were isolated using the Preserved Blood RNA Purification Kit I (Norgen Biotek Corp., Ontario, Canada). RNA purity and concentration were determined using a NanoDrop ND-1000 spectrophotometer (Thermo Fisher), and RNA integrity was assessed using a Bioanalyzer 2100 (Agilent Technologies, Santa Clara, CA, USA). All the 22 RNA samples were processed. The transcriptome microarray data production, pre-processing, and analysis are depicted in Mach et al.[25].

Transcripte profiling was performed using an Agilent 4X44K horse custom microarray (Agilent Technologies, AMADID 044466). All of the steps are detailed here[73,74]. We refer to our previous work for more details on the pre-processing, normalization, and application of linear models[25]. Given our interest in understanding the role played by mitochondria during exercise, the set of 801 differentially expressed mitochondrial genes reported by our team[25] was selected for the downstream steps of analysis (Supplementary Data 15).

**Proton magnetic resonance (¹H NMR) metabolite analysis in plasma**. As described elsewhere[24,70], the plasma metabolic phenotype of endurance horses was obtained from ¹H NMR spectra at 600 MHz. The ¹H NMR spectra were acquired at 500 MHz with an AVANCE III (Bruker, Wissembourg, France) equipped with a 5 mm reversed QXI Z-gradient high-resolution probe. Further details on sample preparation, data acquisition, quality control, spectroscopic data pre-processing, and data pre-processing, including bin alignment, normalization, scaling, and centering, are broadly discussed elsewhere[75]. Details on metabolite identification are described in our previous work[24,25].

**Biochemical assay data production**. Sera were assayed for total bilirubin, conjugated bilirubin, total protein, creatinine, creatine kinase, β-hydroxybutyrate, and aspartate transaminase (ASAT), γ-glutamyltransferase and serum amyloid A levels on an RX Imola analyzer (Randox, Crumlin, UK).

**Blood acylcarnitine profiling**. As a proxy for mitochondrial β-oxidation, the serum acylcarnitine profiles were produced and analyzed as described elsewhere[6]. In the positive mode, free carnitine and 27 acylcarnitines were analyzed for their butyl ester derivatives by electrospray tandem mass spectrometry (ESI-MS-MS) on a triple quadrupole mass spectrometer (Xevo TQ-S Waters, Milford, MA, USA) using deuterated water.

**Fecal measurements: SCFA, DNA extraction, and microorganism concentrations**. Fresh fecal samples were obtained while monitoring the horses before the race. One fecal sample from each animal was collected immediately after defecation[24,76], and three aliquots (200 mg) were prepared. The dehydration experienced by most horses after the race altered intestinal motility and feces shedding, making it impossible to recover the feces immediately after the race.

Aliquots for SCFA analysis and DNA extraction were snap-frozen.

SCFA levels were determined by gas chromatography using the method described elsewhere[77].

Total DNA from the 11 samples was extracted from ~200 mg of fecal material using the EZNA Stool DNA Kit (Omega Bio-Tek, Norcross, Georgia, USA) following the manufacturer's instructions. DNA was then quantified using a Qubit and a dsDNA HS assay kit (Thermo Fisher).

As detailed in our previous studies[24,25], concentrations of bacteria, anaerobic fungi, and protozoa in fecal samples were quantified by qPCR using a QuantStudio 12K Flex platform (Thermo Fisher Scientific, Waltham, USA). Primers for real-time amplification of bacteria (FOR: 5′-CAGCMGCCGCGGTAANWC-3′; REV: 5′-CCGTCAATTCMTTTRAGTTT-3′), anaerobic fungi (FOR: 5′-TCCTTAC CCTTTGTGAATTTG-3′; REV: 5′-CTGCGTTCTTCATCGTTGCG-3′) and protozoa (FOR: 5′-GCTTTCGWTGGGTAGTGTATT-3′; REV: 5′-CTTGCCCTCYAATCGTWCT-3′). Details of standard dilutions series, the thermal cycling conditions, and the estimation of the number of copies are detailed elsewhere[24,25].

**Fecal microbiota: V3–V4 16S rRNA gene sequencing and data pre-processing**. A detailed description of the DNA isolation process, V3–V4 16S rRNA gene sequencing-PCR amplification, is presented by our group[19,20,24,25,76,78,79]. A negative control sample alongside biological samples at the DNA extraction and PCR steps was considered in attempts to control DNA contamination before and after sequencing. In addition, contamination was minimized through laboratory techniques such as UV irradiation of material, ultrapure water, the DNA-free Taq DNA polymerase, and the separation of pre-and post-PCR areas.

The Divisive Amplicon Denoising Algorithm (DADA) was implemented using the DADA2 plug-in for QIIME 2 (v.2021.2) to perform quality filtering and chimera removal and to construct a feature table consisting of read abundance per amplicon sequence variant (ASV) by sample[80]. Taxonomic assignments were given to ASVs by importing Greengenes 16S rRNA Database (release 13.8) to QIIME 2 and classifying representative ASVs using the naive Bayes classifier plug-in[81]. The phyloseq (v.1.36.0)[82], vegan (v.2.5.7)[83], and microbiome (v.1.14.0) packages were used in R (v.4.1.0) for the downstream steps of analysis. A total of 364,026 high-quality sequence reads were recovered for the 11 horses of the study (mean per subject: 33,093 ± 17,437, range: 12,052–62,670). Reads were clustered into 5412 chimera- and singleton-filtered ASVs at 99% sequence similarity. The genera taxonomic assignments and counts for each individual are presented in Supplementary Data 10.

The negative control sample did not yield a band on the agarose gel, and the concentration of the purified amplicon was undetectable (<1 ng/µL). Nevertheless, the decontam (v.1.14.0) R package was used to identify and visualize possible contaminating DNA features in the negative control sample. The function isContaminnat() was used to determine the distribution of the frequency of each contaminant feature as a function of the input DNA concentration. Only 6 ASV were statistically classified (p < 0.05) as contaminants, although their frequency plots showed they were non-contaminants (Supplementary Fig. 11).

**Fecal metagenome: Shotgun sequencing data production and analysis**. Metagenomic sequencing was performed using the same DNA extractions. For each individual, a paired-end metagenomic library was prepared from 100 ng of DNA using the DNA PCR-free Library Prep Kit (Illumina, San Diego, CA, USA). The size was selected at about 400 bp. The pooled indexed library was sequenced in an Illumina HiSeq3000 using a paired-end read length of 2 × 150 pb with the Illumina HiSeq3000 Reagent Kits at the PLaGe facility (INRAe, Toulouse).

**MAG assembly and annotation**. Raw metagenomic reads were quality-trimmed, assembled, binned, and annotated using the ATLAS pipeline, v.2.4.4[84]. In short, using tools from the BBmap suite v.37.99[85], reads were quality trimmed with ATLAS parameters: preprocess_minimum_base_quality = 10, preprocess_minimum_passing_read_length = 51, preprocess_minimum_base_frequency = 0.05, preprocess_adapter_min_k = 8, preprocess_allowable_kmer_mismatches = 1, and the preprocess_reference_kmer_match_length = 27. The contamination from the horse genome (available at NCBI sequence archive with the accession number GCA_002863925.1; Equus_caballus.EquCab3.0) was filtered out using the following settings: contaminant_max_indel = 20, contaminant_min_ratio = 0.65,

contaminant_kmer_length=13, contaminant_minimum_hits = 1, and con-taminant_ambiguous=best. Reads were error corrected and merged before assembly with metaSPAdes v.3.13.1[86] with the subsequent parameters: spa-des_k=auto, prefilter_minimum_contig_length = 300, minimum_average_coverage = 1, minimum_percent_covered_bases=20, and minimum_contig_length = 500 after filtering. QUAST 5.0.2[87] was used to evaluate the quality of each sample assembly. Since a high diversity between individuals was described through 16 S rRNA amplicon analysis, we first assembled each sample independently. Contigs from single samples were clustered into metagenomic bins using MetaBAT 2 (v.2.14)[88] with the following parameters: sensitivity=sensitive, min_conti-g_length = 1500 and Maxbin 2.0 v.2.2.7[89] with the parameters set to max_itera-tion = 50, prob_threshold = 0.9, and min_contig_length = 1000. Contig predictions were combined using DAS Tool v.1.1.2-1[90] with diamond engine and score_threshold set to 0.5.

ATLAS configuration file, summaries of individual samples quality control, contigs from the individuals, and detected bins are available at the INRAE data repository (https://doi.org/10.15454/NGBSPC)[91] and are contained in the files "ATLAS_config.yalm", "ATLAS_dag.pdf", "notebook.html", "ATLAS_QC_report.html", and "ATLAS_bin_report_DASTool.html".

Assembly statistics for the predicted MAGs such as completeness, redundancy, size, number of contigs, contig N50, length of the longest contig, average GC content, and the number of predicted genes were computed using the lineage workflow from CheckM v.1.1.3[92]. MAGs were designated as near-complete drafts if they had completeness ≥90%, redundancy <5%, transfer RNA gene sequences for at least 18 unique amino acids, or medium-quality drafts if they had completeness ≥50% and a redundancy <10%. A summary of the assembly statistics for the predicted MAGs is available at the INRAE data repository: https://doi.org/10.15454/NGBSPC[91] as "ATLAS_assembly_report.htlm".

Because the same MAG may be identified in multiple samples, dRep v.2.2.2[93] was used to obtain a non-redundant set of MAGs by clustering genomes to a defined average nucleotide identity (ANI) and returning the representative with the highest dRep score in each cluster. The parameters used were set to ANI = 0.95, overlap=0.6, length=5000, completeness=50, contamination=10, and N50 = 0.5. Only the highest-scoring MAG from each secondary cluster was retained as the winning genome in the dereplicated set. The abundance of each MAG was then quantified across samples by mapping the reads to the non-redundant MAGs using the BBmap suite v.37.99[85] (pairlen = 100, minid=0.9, mdtag=t, xstag=fs, nmtag=t, sam=1.3, ambiguous=best, secondary=t, saa=f, maxsites=10). The sample-specific median coverage of each MAG was then computed using pileup within BBMap with default parameters.

For the taxonomic annotation, ATLAS predicted the genes of each MAG sequence using Prodigal v.2.6.3[94] with single-mode and closed-end parameters. The taxonomy of the predicted MAGs was inferred using the genome taxonomy database (GTDB-Tk)[43] (v.5.0, release 95 (July 17, 2020)). As such, GTDB-Tk taxonomy names were used throughout this paper. In addition, domain-specific trees incorporating the predicted MAGs were inferred by constructing a maximum-likelihood tree using the de novo workflow in GTDB-Tk v.5.0 with the following parameters: --bacteria | --archaea, min_perc_aa=50, prot_model=WAG. Trees were visualized using ggtree (v.3.0.2) in the R package.

To assess the contribution of the constructed MAGs to the functional potential of the gut microbiome, the predicted gene and proteins extracted by Prodigal during the CheckM pipeline were compared to the EggNOG database 5.0 using eggnog-mapper (v2.0.1). KEGG annotation (Kyoto Encyclopedia of Genes and Genomes) and CAZymes annotation (Carbohydrate-active Enzyme) were extracted from this output. Since the detection of KOs and CAZymes families is likely influenced by sequencing depth, their relative abundance was normalized to the abundance of the MAG they derived from. Pathways attributed to each KO were annotated from the KEGG database (downloaded 23-October-2021; https://www.genome.jp/brite/ko00001).

The uniqueness of our predicted MAG catalog was confirmed by dereplicating them with the 121 MAGs produced by Gilroy et al.[44] and three reported by Youngblut et al.[45] using dRep v.3.2.0[93] with parameters: P_ani = 0.9, S_algorithm 'ANImf', S_ani = 0.99, clusterAlg 'average', cov_thresh = 0.1, coverage_method 'larger.' dRep performed pairwise genomic comparisons by sequentially applying an estimation of genome distance and an accurate measure of average nucleotide identity. Visualizing and comparing highly similar genomes were performed using the CGView family of tools (http://wishart.biology.ualberta.ca/cgview/).

**Construction of the integrated gene catalog**. The establishment and assessment of the quality and representation of the microbiome gene catalog were performed through the metagenomic ATLAS pipeline (v.2.4.4)[84]. As described above, we first assembled the clean reads into longer contigs.

Genes were predicted by Prodigal v.2.6.3 and then clustered using linclust[95] to generate a non-redundant gene catalog. Redundant genes were removed with linclust using the following parameters: minlength_nt = 100, minid = 0.95, coverag = 0.9, and subsetsize = 500,000. The quantification of genes per sample was done through the combine_gene_coverages() function in the ATLAS workflow, which aligned the high-quality clean reads to the gene catalog using the BBmap suite v.37.99[85] (minid = 0.95, mdtag = t, xstag = fs, nmtag = t, sam= 1.3, ambiguous = all, secondary = t, saa = f, maxsites = 4). Taxonomic and function

annotations were done based on the EggNOG database 5.0 using eggnog-mapper (v.2.0.1) (emapper.py–annotate_hits_table {input.seed}–no_file_comments). The eggNOG numbers corresponding to CAZymes based on homology searches to the CAZyme database were retrieved from these. We used the derived eggNOG abundance matrix to obtain a CAZyme profile per sample. Similarly, KEGG annotation was recovered from the EggNOG output. KEGG gene IDs were mapped to KEGG KOs and used to get the KEGG functional pathway hierarchy. Furthermore, using mmseqs2 (v.13.45111) to find genes at a 95% similarity threshold and 80% overlap, we compared our gene catalog with a previously published gene catalog containing ~4 million genes[30]. The parameters used were the following: easy-search --search-type 3 --min-seq-id 0.95 --cov-mode 0 -c 0.8 --threads 16 --alignment-mode 3 --max-seq-len 100000.

The annotated gene catalog fasta file is deposited at DDBJ/ENA/GenBank Whole Genome Shotgun under the BioProject ID PRJNA438436 and is also available at https://doi.org/10.15454/NGBSPC[91] as "Genecatalog_with-note.fna.gz". The KO and CAZymes derived from the gene catalog are available in the same INRAE data repository and are in the "Genecatalog_KO.tab" and "Genecatalog_CAZy.tab" files, respectively.

**Annotation of metagenomes using Kaiju**. The kmer-based kaiju v.1.8.0 (https://github.com/bioinformatics-centre/kaiju)[31] approach was used for microbial taxo-nomic profiling of the trimmed shotgun metagenomes and the microbial gene catalog. The microbial gene catalog fasta, core group genes fasta, and paired reads after quality trimmed and decontamination from the horse genome were used and annotated against the NCBI nr_euk reference database (released on May 25, 2020) containing all proteins belonging to archaea, bacteria, fungi, microbial eukaryotes, and viruses for classification in Greedy run mode with -a greedy -e 3 allowing for maximum three mismatches. By default, Kaiju returned a "NA" if it could not find a taxonomic classification at certain ranks. The Kaiju's tab-separated output files were imported into Krona and converted into HTML files. They are available at https://doi.org/10.15454/NGBSPC)[91] under raw-samples.nr_euk.kaiju.html.

**Dominant phylotypes**. To circumvent the problem of false-positive species pre-dictions due to misalignment and contamination, we defined an abundance threshold of 25%, where the top 25% abundant species in at least 50% of the individuals were retained using the filterfun_sample() function in the phyloseq R package. This reduced background noise but kept information on poorly-described species if they were ubiquitously found in the samples. The dominant phylotypes abundance, taxonomy, and the associated metadata are available at https://doi.org/10.15454/NGBSPC as "Ecaomic_dominant_phylotypes_nonrariefied.rds".

**Resistome**. The high-quality clean paired reads were aligned to the ResFinder database (accessed March 2018, v.4.0) using bowtie2 (v.2.3.5). ResFinder is a manually curated database of horizontally acquired antimicrobial resistance (AMR) genes. It contains many genes with numerous highly similar alleles (e.g., β-lacta-mases). To avoid random assignment of read pairs on these high-identity alleles, the database was clustered at 95% of identity level, over 200 bp using CDHIT-EST (options -G 0 -A 200 -d 0 -c 0.95 -T 6 -g 1)[96] and a reference sequence was attributed to each cluster. Two successive mappings were done: (i) the first map-ping with standard parameters (bowtie2 --end-to-end --no-discordant --no-overlap --no-dovetail –no-unal) on the complete ResFinder database, and (ii) a second mapping on the clustered database using the reads from the first mapping, with less stringent parameters (bowtie2 --local --score-min L,10,0.8). More than 99% of the reads from the first mapping correctly aligned on a cluster reference sequence in the second mapping.

Counts from the second mapping were normalized by computing the RPKM (reads per kilobase reference per million bacterial reads) value for each ResFinder reference sequence. The RPKM values were calculated by dividing the mapping count on each reference by its gene length and the total number of bacterial read pairs for the samples and multiplying by $10^9$. A minimum of 20 mapped reads was considered to validate the presence of an AMR gene cluster.

**Statistics and reproducibility**

*Biodiversity and richness analysis: α- and β-diversity*. The microbiome R package allowed us to study global indicators of the gut ecosystem state, including measures of evenness, dominance, divergences, and abundance. Comparison of the gut α-diversity indices between groups was performed by a two-sided Wilcoxon rank-sum test (pairwise comparison). Benjamini–Hochberg multiple testing correction $p < 0.05$ was set as the significance threshold for comparison between groups.

To estimate β-diversity, Bray–Curtis dissimilarity was calculated using the phyloseq R package. All samples were normalized using the rarefy_even_depth() function in the phyloseq R package, which is implemented as an ad hoc means to normalize features resulting from libraries of widely differing sizes. The PerMANOVA test (a non-parametric method of multivariate analysis of variance based on pairwise distances) was implemented using the adonis() function in the vegan R package and the pairwise.Adonis2() function from the pairwiseAdonis (v.0.4) R package tests the global association between ecological or functional community structure and groups. The model was adjusted by factors affecting the microbiome: age, sex, and dietary macronutrient intake.

*The core microbiome*. The core group of genes in the catalog was defined as the genes present in all individuals.

The dominant core microbiome at the genus level was calculated using a detection threshold of 0.1% and a prevalence threshold of 95% in the microbiome R package.

*Inference and analysis of SPIEC-EASI microbiome networks*. The SParse InversE Covariance Estimation for Ecological Association Inference method (SPIEC-EASI)[97] was used to identify sub-populations (modules) of co-abundance and co-exclusion relationships between dominant phylotypes and CAZy classes abundances matrices. Specifically, the method allows microorganisms and functions to interact differently, from bidirectional competition to mutualism or not interacting at all. The statistical method SPIEC-EASI comprises two steps: a transformation for compositionality correction of the feature matrices and estimation of the interaction graph from the transformed data using sparse inverse covariance selection. The sparse graphical modeling framework was constructed using the spiec.easi() function of the SpiecEasi package (v.1.1.1). The features were clustered using the method = mb, lambda.min.ratio = 1e$^{-5}$, nlambda = 100, pulsar.params=list (thresh = 0.001). Regression coefficients from the SPIEC-EASI output were extracted and used as edge weights to generate a feature co-occurrence network R igraph package (v.1.2.6) and Cytoscape (v.3.8.2).

*Integrative statistical analysis*. Data integration was carried out using several approaches and different combinations of datasets. Before the integration, we applied some additional pre-processing steps to our exploratory datasets. In particular, to eliminate intra-individual variability and focus on the differential signals between T1 and T0, we considered Δ values (T1–T0) for each of these datasets, namely biochemical assay data and metabolome acylcarnitine profiles, and gene expression data. For the transcriptome, we constructed a matrix of log-transformed expression values between T1 and T0 (e.g., the difference in $\log_2$-normalized expression between T1 and T0).

The integration of data was then performed using complementary methods and working with different datasets available, namely: (1) Δ values of mitochondrial-related genes; (2) Δ values of $^1$H NMR metabolites; (3) Δ values of the biochemical assay metabolites; (4) Δ values of plasmatic acylcarnitines; (5) the fecal SCFAs at T0; (6) the bacterial, ciliate protozoa and fungal loads at T0; (7) the dominant gut phylotypes at T0; (8) the CAZymes profiles at T0; (9) the KOs at T0, and the (10) athletic performance data.

As a first integration approach, a global non-metric multidimensional scaling (NMDS) ordination was used to extract and summarize the variation in microbiome composition using the metaMDS() function in the vegan R package. Stress values were calculated to determine the number of dimensions for each NMDS.

The explanatory datasets were then fit to the ordination plots using the envfit() function in the vegan R package[98] with 10,000 permutations. Each covariate's effect size and significance were determined, and all *p*-values derived from the envfit() function were adjusted Benjamini–Hochberg. Variation partitioning was performed using the varpart() function in vegan in R.

The N-integration algorithm DIABLO of the mixOmics R package (http://mixomics.org/, v6.12.2) was used as a second integrative approach. It is to be noted that, in the case of the N-integration algorithm DIABLO, the variables of all the datasets were also centered and scaled to unit variance before integration. In this case, the relationships among all datasets were studied by adding a different categorical variable, e.g., the cardiovascular fitness of horses. Horses with poor cardiovascular fitness (*n* = 8) were compared to horses with enhanced cardiovascular fitness (*n* = 3). DIABLO seeks to estimate latent components by modeling and maximizing the correlation between pairs of pre-specified datasets to unravel similar functional relationships[99]. To predict the number of latent components and the number of discriminants, the block.splsda() function was used. The model was first fine-tuned using leave-one-out cross-validation by splitting the data into training and testing. Then, classification error rates were calculated using balanced error rates (BERs) between the predicted latent variables with the centroid of the class labels using the max. dist() function.

Finally, the DESeq2 (v.1.32.0)[100] R package was used to test differential abundances analysis between groups for the dominant phylotypes, MAGs, and the genetic functionalities derived from KOs and CAZymes at the basal time. DESeq2 assumes counts can be modeled as a negative binomial distribution with a mean parameter, allowing for size factors and a dispersion parameter. The *p*-values were adjusted for multiple testing using the Benjamini–Hochberg procedure. DESeq2 comparisons were run with the parameters fitType="parametric" and sfType="Wald".

*The validation cohort*. The validation set consisted of 22 pure-breed or half-breed Arabian horses (12 females, three males, and seven geldings; age: 9.2 ± 1.27) not included in the experimental set to ensure that the observed effects were reproducible in a broader context (Supplementary Data 20). Five animals were enrolled in a 160 km endurance competition among the horses in the validation set, while 17 were in a 120 km race. The management practices throughout the endurance ride and the International Equestrian Federation (FEI) compulsory examinations and the weather conditions, terrain difficulty, and altitude were that of the

experimental set. All the participants enrolled in the study (experimental and validation set) competed in the same event in October 2015 in Fontainebleau (France). The cardiovascular capacity was created as described in the "Performance measurement" section as a composite of post-exercise heart rate, cardiac recovery time, and average speed during the race. Then, the HIGH, MEDIUM, and LOW groups were determined according to the interquartile range of the composite cardiovascular fitness values. HIGH included individuals with cardiovascular fitness values above the 75$^{th}$ percentile, LOW below the 25$^{th}$ percentile, and MEDIUM, the individuals ranging in between.

The PerMANOVA test was implemented by using pairwise.Adonis2() function from the pairwiseAdonis R package. The model was adjusted by factors affecting the microbiome: age and sex. The homogeneity of group dispersions (variance) was applied via the betadisp() function of the vegan package to account for the confounding dispersion effect. The one-way ANOVA with Tukey's honest significant differences (HSD) method for pairwise comparisons was performed using the TukeyHSD() function in the stats R package (v.3.6.2).

The PLS-DA was used to identify the key genera responsible for the differences in the groups using the mixOmics[101] R package (v. 6.18.1). In addition, as PLS-DA loadings may be misleading with highly correlated variables, the differences in each relative genus' abundance between the groups were quantified by DESeq2 R package.

**Reporting summary**. Further information on research design is available in the Nature Research Reporting Summary linked to this article.

## Data availability

The datasets presented in this study can be found in different online repositories. Microarray expression data (MIAME compliant) are available in the Gene Expression Omnibus (GEO) repository under the accession number GSE163767. Metabolomic data are available in the NIH Common Fund's Data Repository and Coordinating Center UrqK1489 (http://dev.metabolomicsworkbench.org:22222/data/DRCCMetadata.php?Mode=Study&StudyID=ST000945). The gut metagenome 16S rRNA targeted locus data from the experimental and validation sets are available in the DDBJ/EMBL/GenBank under the BioProject PRJNA438436. The accession numbers of the BioSamples included in the experimental set are SAMN08715729, SAMN08715728, SAMN08715727, SAMN08715725, SAMN08715723, SAMN08715721, SAMN08715719, SAMN08715718, SAMN08715714, SAMN08715713, SAMN08715710. The SRR accession numbers for the 16S rRNA targeted locus data are: SRR13664931, SRR13664928, SRR13664927, SRR13664925, SRR13664924, SRR13664923, SRR13664921, SRR13664919, SRR13664918, SRR13664917 and SRR13664916, respectively. Moreover, the raw metagenomic sequence data of the 11 athletes have been deposited at DDBJ/ENA/GenBank Whole Genome Shotgun under the same BioProject ID PRJNA438436 and BioSamples numbers. The accession numbers ranged from SRR17543914 to SRR17543904. All metagenome assemblies and sequences of MAGs have also been deposited at DDBJ/ENA/GenBank Whole Genome Shotgun under the same BioProject ID PRJNA438436, with the genome accession numbers ranging from JAKSHS000000000 to JAKSVZ000000000. They are also available at the INRAE institutional data repository powered by Dataverse with https://doi.org/10.15454/NGBSPC[91]. Last, the horse gut microbiome gene catalog is available at DDBJ/ENA/GenBank under the same BioProject ID PRJNA438436 and the accession numbers from JALNLV000000000 to JALNLY000000000. The catalog is also available at the INRAE institutional data repository powered by Dataverse with https://doi.org/10.15454/NGBSPC[91]. Datasets generated or analyzed during the study are included in this published article as Supplementary Data. Other data supporting this study's findings are available in the INRAE institutional data repository powered by Dataverse with https://doi.org/10.15454/NGBSPC[91]. They have been appropriately specified in the text where required. The source data underlying the graphs and charts presented in the main figures are stored in Supplementary Data 23. It contains a single experiment-level phyloseq object with the ASV matrix, all related phylogenetic sequencing data, annotation, and metadata. All other data are available from the corresponding author on reasonable request.

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

## Acknowledgements

We are grateful to Marine Beinat, Julie Rivière, Emmanuelle Rebours, Jordi Estellé, Caroline Morgenthaler, and Arni Janssen for participating in the sample collection and organization during the project. We particularly thank Valentine Ballan and Fanny Blanc, who helped isolate the blood RNA. We also thank Catherine Philippe for the SCFA analysis in feces, and Diane Esquerré, who prepared the libraries and performed the MiSeq sequencing at the GeT-PlaGe genomics core facility (INRAE, Toulouse, https://get.genotoul.fr/) and Estel Blasi, who created the picture of horses. Last, we are grateful to all the horse owners, riders, and endurance race organizers who participated in the study. This work was funded by grants from the *Fonds Eperon*, the *Institut Français du Cheval et de l'Equitation* (IFCE), the *Association du Cheval Arabe* (ACA), and the Animal Genetic Department at *Institute National de la Recherche pour l'Agriculture, l'alimentation et l'environnement* (INRAE).

## Author contributions

N.M., C.R., and E.B. conceived the study. N.M. designed and performed the 16S rRNA bioinformatics data analyses. C.M. and O.R. optimized the ATLAS pipeline to create the gene catalog and MAGs repertoire and run kaiju. S.L. analyzed the resistome. N.M. designed and performed all statistical analyses and made all figures and tables. S.P. prepared the metagenome sequencing libraries. L.L.M. performed the metabolomic experiment and explored the metabolite peaks. C.R. and E.B. managed the GenEndurance project and organized the sample collection during the race. N.M. and G.S. wrote the manuscript. The manuscript was revised by G.S., E.B., C.R., C.M., S.L., and L.L.M. All authors read and approved the final manuscript.

## Competing interests

The authors declare no competing interests.
