## [Peer Review File · Communications Biology]

Reviewers' comments:

Reviewer #1 (Remarks to the Author):

I have reviewed the manuscript entitled "The first horse gut microbiome gene catalog reveals that rare microbiome ensures better cardiovascular fitness during endurance racing". The authors analyzed combined microbial and host omic datasets and report an associative link between horse endurance capability and its microbiome gene function. I feel that this is an excellent body of work that provides evidence of biotic interactions and functional gene potential in the gut of horse athletes. However, I have several comments that I feel should be addressed prior to publication.

Line 29: Replace "unrevealed" with "revealed"

Line 97: Replace "host sequences" with "host sequence"

Line 98: "Replace "millions" with "million"

Line 100: Replace "millions of paired" with "million paired"

Line 116: Replace "phylum" with "phyla"

Line 129: Replace "All of them" with "All of "these"

Line 174-179: This should go in the Discussion section and be followed by something like, "For example, our finding of glycoside hydrolase, polysaccharide lyases, and indicate... It has been reported that... Therefore...)"

Line 189: The performance level of this cluster should be noted here.

Line 201: I suggest using an Oxford comma here and throughout the manuscript.

Line: 226: What was the model? adonis2(Bray~fitness)? What was the effect of sex and horse breed? Were horse ID (block/strata=Horse ID) and sampling time (T0/T1) considered? Alternatively, perform PERMANOVA for T0 and T1 separately. Please report R2.

Line 228: Replace "players" with "taxa"

Line: 229: Firmicutes is a large group. Can you additionally report differentially abundant genera here?

Line 312: Replace "recalled" with a different word

Line: 313: Replace "commensals" with a different word

Line: 322: "Ecosystems" is an overreach. Limit the interpretation to horses

Line 358: Replace "be likely" with "likely be"

Line 366: I suggest tempering this sentence. Replace "highlighted with "indicated". Replace "one of the most effective ways to modulate" with "a potentially highly effective way to modulate".

Line 399: Replace "this data" with "these data"

Line 429: Were any of these horses closely related?

Line 563: Please provide any additional settings/options used for BBmap, metaSpades, and other scripts. Readers greatly appreciate this.

Line 583: Was BBmap used here? Were the settings these same as was used for removing horse DNA?

Line 665: Please see my comment for line 226

Line 736-746: Sampling time must also be included in the model or tests should be performed for T0 and T1 separately.'

Reviewer #2 (Remarks to the Author):

Mach and collaborators present a functional survey of the microbiome of horse athletes. Through pathway and genome reconstruction and analyses, they suggest clustering of microbiome profiles based on the dominance of Lachnospiraceae or the presence of rare species. They suggest the presence of rare species associates with improved cardiovascular capacity. The manuscript is well written, and the data great number of data analyses presented are robust for the most part. However, some info appears to be rather strong and speculative, such as the contentions of rare taxa contributing to enhanced athletic performance. In this regard, there is also lack of info on the nature of these rare taxa and the methods used to validate them or discard their origin due to possible contaminating sources, as there is indication of the use of negative controls.

Specific comments are as follow:

Title is speculative, the word "ensures" is rather strong, as the data mainly show associations

Abstract:

-L33: what does the article "they" refer to? The rare species? Also, can authors include info on what they consider rare (% threshold) and some examples of these rare taxa?

-L34: Can the authors briefly elaborate on what specific "mitochondria-mediated mechanisms" they refer to?

-It is not clear if what associated to improved endurance is the presence of rare species or the functions encoded by them.

Intro:

L45: "160 km at 20 km/h" is not clear, do they mean 160km/h?

L57-60: can authors elaborate a bit more on what microbially-produced metabolites are associated to those desirable phenotypic traits? Same for L77-78.

Results

L103-104: Could you please clarify what this statement means? Do you mean that not all horses harbored the ~25M gene catalog, meaning that these are not core genes and therefore were not found across all horses? So is the gene catalog presented not really a "core" gene set?

L107: Please state the database used. Also, if 61% were exact matches, does that mean 39% of reads were unknown? % of unknown/un-annotated reads should be described.

L140: why are authors reporting pathways associated with endocrine system and neurodegenerative diseases? These are eukaryote pathways that show up because of shared prokaryote and eukaryote (human) genes involved in diverse pathways- these pathways, as inferred from bacterial metagenomes should be filtered out.

L156-158 (Figure 1d), suggestion: Perhaps it maybe be more interesting, in the context of the question pursued by authors, to show the CAzyme data (FS2) as main figure, 1d for instance? AMR genes analyses are interesting but unclear what these data brings to understand the effect of the microbiome on athlete performance (appears to be rather distracting since no reference on this is made in the rest of the manuscript). AMR genes are expected to ubiquitous in all microbiomes, regardless of exposure, so it is not clear what the intent is with this analysis.

L199-200: Can authors provide brief description of those macronutrient intake analyses? Up to this point it was not clear these analyses had been made. A brief intro (few lines) stating which analyses, why and how they were conducted ? Also, it is clear that macronutrient intake did not differ between horses in the low and high performance clusters, but why wasn't that factor included in the PERMANOVA analyses validating the microbial clusters? That analysis probably should be included to test for the same clusters, based on CAzymes and K0s.

L229: Shouldn't it read "higher abundance OF"?

Discussion

L336: What is meant by "tissue repair" ? what microbial functions reflect that end?

L357-363: This contention is a bit vague. Although it may seem intuitive as per the results and what authors want to convey, can they elaborate bit more on what these interventions may look like? How will these interventions will "create more space" for rare taxa? What specific probiotics/fibers are they referring to that may expand rare taxa while constraining Lachnospiraceae? How would fecal transplants increase rare taxa and inhibit dominant Lachno? This is may be a vague statement since; 1) Lachnospiraceae are dominate phylotypes in most herbivore mammals, and especially equines 2) authors have not really elaborated or expanded on the true abundance, nature and function of these rare taxa, 3) The number of horses that are deemed as high performers and that contain these rare taxa is only three, 4) it is impossible to validate the role of these taxa based on the 16S rRNA dataset (what rare taxa could be validated in these dataset?)

In this regard, both in the results and discussion sections there should be some lines devoted to describe discuss the abundance (ranges, averages) of these rare taxa and info on the environments they have been normally detected in. There should also be some discussion as to why authors discard the possibility of these taxa being the result of contamination? Why didn't the authors use negative (blank extraction) controls? Addressing these issue is key as blooms in microbial contamination (e.g. kit associated contaminants) can significantly bias alpha diversity, which authors claim to be a marker of high endurance/athletic performance.

L367-368: What microbial pathways and/or genes are specifically involved in mitochondrial biogenesis and energy resilience? This is not clear in the analyses nor supported by existing data.

L373-380: this contention is highly speculative, authors have provided no direct evidence that these are taxa have any capacity to break down host glycans or that the metabolism of such substrates can significantly contribute to the energetic landscape of the host, or to mitochondrial function. If anything one would expect that energy harvest from dietary glycans far exceed any contributions (if any) from metabolizing animal glycans. Also, what metabolic products would be released by these metabolic

route, that are not already released by feed fermentation?

Many of the rare taxa found are involved in methane metabolism. Can authors elaborate on possible contributions of these metabolic pathway, from an energetic stand point, on horse endurance? Especially in the context of lack of differences in SCFA profiles.

L402-406: Does the finding of rare taxa correlates with the remarkable depth of this metagenomic dataset ? In other words, would these rare taxa be detected in any horse population regardless of phenotype?

In this regard, it is not clear how the discriminatory power of rare taxa was validated. So far it seems the prevalence of rare taxa in high performers was assessed via loadings in the MDS projection, but there doesn't seem to be a procedure or test (e.g. via DeSeq, ANCOM, LEfSe) showing how discriminating taxa were validated. It is assumed that figure S8 accomplish this goal, but such figure does not appear to be referenced in the text (apologies if I am missing this)

L411: There are conflicting views regarding the potentially beneficial role of Akkermansia, as its abundance has also been associated with the erosion of the mucus layer due to lack of fermentable fiber:

L414: How can an "intimate" connection between gut microbes and mitochondrial function., at the systemic level, be assessed? Is that possible at all? This contention is highly speculative.

Authors are encouraged to include a study limitation section based on the concerns raised above – including, among others, validation of rare taxa as markers of high performers, inability to assess their true function or provenance, their true (or likely) impact on the systemic landscape of the host and the fact that rare taxa are only observed in three horses.

Methods

L554: Info on negative controls for the microbiome procedure should be included

L670: Where is this interindividual/PAM clustering analyses included and what is its purpose?

Reviewer #3 (Remarks to the Author):

Using elite racehorses as a model, this work sought to build the first gene microbial catalog for this animal in an effort to better elucidate cardiovascular fitness. In line with Communications Biology, this manuscript represents a significant advance in the integration of systems biology and exercise science, presenting novel findings. While quite limited in sample size, the considerations and new evidence this work presents have important implications for future microbiome research in relation to exercise and physical activity (in general). For instance, microbiomes that supported rarer species were able to provide an expanded repertoire of metabolic pathways and mitochondrial potential as compared to those less diverse and higher in abundance of taxa in the Lachnospiraceae family. While not conducted in humans, the use of an equine model is innovative and well suited for the present examination. Moreover, the comparative physiology aspects of this work are commendable and were well received by this reviewer.

Overall, the manuscript is clearly written and of appropriate length. However, in relation to not overselling the thesis of this work, extrapolating findings from this model needs more care. For example, while the introduction generally sets the stage well for the workflow, one critical issue that should be noted is the distinct differences in gastrointestinal structure and physiology of those species

better equipped to endure high-end cardiovascular output. In the spirit of the comparative aspects of this work, clearly horse and human GI tracts have distinctions. Perhaps highlighting the important similarities of successful phenotypes could better improve such comparisons. Insertion could in the paragraph starting at line 70, where Arabian horses are proposed as the in vivo model. This could also be reinforced in the discussion. With these inclusions this work will have done a fair job in the treatment of the previous literature.

The methods are generally well described, transparent, and performed in a sound manner. For the purpose of reproducibility, it would be a plus to have the main analyses available on a platform such as GitHub or even the outputs on Qiita.

Reviewers' comments:

Reviewer #1 (Remarks to the Author):

1. I have reviewed the manuscript entitled “The first horse gut microbiome gene catalog reveals that rare microbiome ensures better cardiovascular fitness during endurance racing”. The authors analyzed combined microbial and host omic datasets and report an associative link between horse endurance capability and its microbiome gene function. I feel that this is an excellent body of work that provides evidence of biotic interactions and functional gene potential in the gut of horse athletes. However, I have several comments that I feel should be addressed prior to publication.

AU: We thank the reviewer for this appreciation and for taking the time and effort to review the manuscript and put the study into context.

2. Line 29: Replace “unrevealed” with “revealed”

3. Line 97: Replace “host sequences” with “host sequence”

4. Line 98: “Replace “millions” with “million”

5. Line100” Replace “millions of paired” with “million paired”

6. Line” 116: Replace “phylum” with “phyla”

7. Line 129: Replace “All of them” with “All of “these”

AU: We have corrected and replaced all typos from statement #2 to statement #7 in the text.

8. Line 174-179: This should go in the Discussion section and be followed by something like, “For example, our finding of glycoside hydrolase, polysaccharide lyases, and ... indicate... It has been reported that... Therefore...)

AU: Following the reviewer’s suggestion, we have moved the L174-179 from the introduction to the discussion and rephrased the whole paragraph in the following way:

“The current study presents a comprehensive horse gut microbiome gene catalog and its association with endurance performance. The more than 25 million non-redundant genes in this catalog have widened 20-fold the number of genera known to reside in the gastrointestinal tract of horses ¹⁻³, uncovering a substantial number of bacteria ($n = 2,927$), archaea ($n = 174$), and Eukaryota ($n = 1,081$). This catalog captured a wide array of functions associated with complex carbohydrate fermentation and functional capacities that enabled more energy extraction from dietary, microbial, and host resources, which likely provided a performance advantage for athletes. For example, genes encoding enzymes for glycosidic-bond cleavage (GHs and PLs) represented the majority (67%) of the CAZyme genes, highlighting the indispensable role of gut microorganisms in glycan metabolism. Intriguingly, 6.38% of GHs and PLs were involved selectively in cleaving endogenous host-related glycosylated proteins and glycans ⁴. It is yet unclear whether these CAZymes correspond simply to the peptidoglycan-processing machinery required for bacterial cell growth and division, or whether peptidoglycan and highly glycosylated proteins represent a significant carbon source for some bacteria or the host after being transformed to acetate ⁵.

Additionally, we have identified a set of 372 MAGs. The number of microbial genes and MAGs will likely increase as more samples are analyzed, as seen in other livestock gut metagenomes ⁶⁻⁹. Nonetheless, the present metagenomic genes and MAGs repertoire is a significant step

forward in explaining the composition and function of the horse gut microbiome, especially in the context of endurance exercise”.

9. Line 189: The performance level of this cluster should be noted here.

AU: Following the reviewer’s comment, we realized that the first version of the manuscript was unkempt and not clear enough. Therefore, we entirely re-worked the sections entitled: “The dominant basal gut phylotypes defined two taxonomical and functional microbial communities” and “Basal dominant gut phylotypes were closely associated with cardiovascular fitness in athletes”. In this regard, the first section explains now how the basal dominant gut phylotypes were segregated into two different taxonomical and functional microbial communities. The second section shows that clusters were not driven by nutrient intake or horse-centered parameters but rather by cardiovascular performance. Therefore, the performance level of each group is depicted in this second section. Precisely, we detailed that “Horses from cluster 1 had significantly higher cardiovascular fitness relative to cluster 2 members ($p = 0.048$, two-sided Wilcoxon rank-sum test, Fig 3i). That did not incur any increases in blood lactate concentration ($p = 0.921$, two-sided Wilcoxon rank-sum test), which is a proxy for glycolytic stress and disturbance in cellular homeostasis ¹⁰”. Additionally, a violin plot representing the cardiovascular fitness of each cluster is depicted in Fig 3i.

10. Line 201: I suggest using an Oxford comma here and throughout the manuscript.

AU: We have used the Oxford comma throughout the manuscript.

11. Line: 226: What was the model? adonis2(Bray~fitness)? What was the effect of sex and horse breed? Were horse ID (block/strata=Horse ID) and sampling time (T0/T1) considered? Alternatively, perform PERMANOVA for T0 and T1 separately. Please report R2.

AU: The model included fitness(composite.heart.binaire), adjusted by the effect of sex, age of the athlete, energy intake (UFC/day), and macronutrient dietary intakes. Please, see below and the new Supplementary Figure 5 for more details. The breed was not included in the model since values were unique for half-Arabian and Shagya-Arabian. In sum, the model states that gut microbiome structure was mainly independent of age, sex, and diet.

```
pairwise_adonis_results <- pairwise.adonis2(bray ~composite.heart.binaire+ Age + Sexe + UFC_day
+Protein_day+Fiber_day+Fat_day+Non_neutral_detergent_fiber_carbohydrates_day,
data = metadata)
```

```
pairwise_adonis_results
$parent call
[1] "bray ~ composite.heart.binaire + Age + Sexe + UFC_day + Protein_day + Fiber_day + Fat_day +
Non_neutral_detergent_fiber_carbohydrates_day , strata = Null , permutations 999"
```

```
$H vs L
      Df SumOfSqs  R2  F Pr(>F)
composite.heart.binaire 1 0.033201 0.37163 9.3194 0.008991 **
Age                    1 0.007417 0.08303 2.0820 0.219780
Sexe                   2 0.020097 0.22495 2.8206 0.142857
UFC_day                1 0.005502 0.06158 1.5443 0.307692
Protein_day            1 0.004184 0.04683 1.1743 0.432567
Fiber_day              1 0.004027 0.04507 1.1303 0.432567
Fat_day                1 0.004738 0.05303 1.3299 0.372627
Non_neutral_detergent_fiber_carbohydrates_day 1 0.006612 0.07400 1.8558 0.266733
Residual               1 0.003563 0.03988
```

Total 10 0.089340 1.00000

 Signif. codes: 0 '***' 0.001 '**' 0.01 '*' 0.05 '.' 0.1 ' ' 1

Figure S5 – The compositional and functional differences between microbial clusters were not driven by nutrient intake or any horse-centered parameter

(a-g) Non-metric multidimensional scaling (NMDS) ordination plot with Bray-Curtis distance and based on the dominant phototype table. Samples are colored by the amounts of daily fiber, hydrolyzable carbohydrates, protein, fat intake, sex, age (years), and breed, respectively.

Strata were set to null because we did not have repeated fecal samples for the individuals. In this regard, the reviewer’s comments on the longitudinal nature of the data (block/strata=Horse ID; sampling time (T0/T1) revealed that the methods section describing the sample collection was also unclear. We have now explained that due to the particular conditions of field sampling (e.g., significantly delayed gastrointestinal emptying due to dehydration), feces collection after the endurance race was impossible. Therefore, all fecal-related parameters (e.g., the fecal SCFAs, the bacterial, ciliate protozoa, fungal loads, and sequencing data) were obtained only at the basal time (T0). Given that no immediate changes in gut microbiota composition are expected following intense acute exercise in the horse¹¹, we are confident that using T0 data alone for the fecal data is meaningful.

On the other hand, biochemical assay data, metabolome data, acylcarnitine profiles, and gene expression data were collected before (T0) and after the endurance race (T1).

Please see the updated text on the pages 12 (L563-579) and 13 (L620-622).

12. Line 228: Replace “players” with “taxa”

AU: We have replaced players with taxa.

13. Line: 229: Firmicutes is a large group. Can you additionally report differentially abundant genera here?

AU: Yes, we can report differentially abundant genera here. We have re-phrased the text while

reporting differentially abundant genera in each cluster. The new text is: “Cluster 1 mainly included the upregulation of many poorly-characterized taxa ($n = 318$; IQR: $2.90e^{-04}$ - $1.16e^{-04}$; Supplementary Fig 3a-b) that encompassed Proteobacteria ($n = 132$), Actinobacteria ($n = 49$), Verrucomicrobia ($n = 17$), Planctomycetes ($n = 17$), and Cyanobacteria ($n = 7$). Methanogens previously described in the bovine rumen ^{12,13} - including those belonging to the Methanomicrobiales (*Methanotherix*), Methanobacteriales (*Methanosphaera*, *Methanobacterium*, *Methanothermobacter*), and Thermococcales (*Thermococcus*) orders – were also more abundant in that cluster (DESeq2, adjusted $p < 0.05$; Supplementary Table 12 and Supplementary Fig 3c). In contrast to the overwhelming diversity observed in cluster 1, a dwarfed biodiversity defined cluster 2 (Supplementary Table 12 and Supplementary Fig 3a) that assembled around highly efficient fiber degraders (members from the *Lachnospiraceae* taxa: *Anaerostipes*, *Butyrivibrio*, *Blautia*, *Coprococcus*, *Dorea*, *Eubacterium*, *Hespellia*, *Lachnospira*, *Oribacterium*, *Roseburia*, and *L-Ruminococcus*) and downstream users of degradation products (*Prevotella*, and *Treponema*) ¹³ (DESeq2, adjusted $p < 0.05$; Supplementary Table 12 and Supplementary Fig 3c)”. Please, see page 5, L219-232.

14. Line 312: Replace “recalled” with a different word

15. Line: 313: Replace “commensals” with a different word

16. Line: 322: “Ecosystems” is an overreach. Limit the interpretation to horses

17. Line 358: Replace “be likely” with “likely be”

18. Line 366: I suggest tempering this sentence. Replace “highlighted with “indicated”. Replace “one of the most effective ways to modulate” with “a potentially highly effective way to modulate”.

19. Line 399: Replace “this data” with “these data”

AU: We have corrected and replaced all suggested words indicated in statements #14-15,17,19. Additionally, we have tempered the sentences pointed out in comments #16 and #18.

20. Line 429: Were any of these horses closely related

AU: To address this point in the manuscript, we have estimated the pedigree kinship matrix. The kinship2 R package was used to calculate the pedigree kinship matrix of all individual pairs, plot the pedigree, and trim the pedigree object. The kinship coefficient for any two subjects was calculated as the probability that an allele chosen at random for both subjects at a given locus is identical-by-descent, inherited from a common ancestor. The pedigree was calculated using six generations back for the 11 horses of the study. The pedigree kinship matrix was then visualized using the “plot_popkin” function from the popkin R package. The “inbr_diag” function was used to modify the kinship matrix, with inbreeding coefficients along the diagonal, preserving column and row names. This information has been added in the methods section (Please see page 12, L553-561) and Results (Please see page 6: L276-277)

Additionally, we have created Supplementary figure 6 depicting the pedigree and the kinship coefficient matrix. Results demonstrated that clusters 1 and 2 did not overlap with the horse pedigree kinship matrix.

Figure S6 - Genetic relationship of the 11 endurance horses in the experiment

(a) Illustration of a six-generation pedigree plot, with different shapes for male (squares) and female (circles); (b) Heatmap of the kinship coefficient matrix, which assesses the genetic resemblance between horses. Each entry in the matrix is the kinship coefficient between two

subjects. Animals are arranged in the order of their genetic relatedness. Genetically similar animals are near each other. In the heatmap, red = high values of genetic relatedness, and white = low values of genetic relatedness.

21. Line 563: Please provide any additional settings/options used for BBmap, metaSpades, and other scripts. Readers greatly appreciate this.

AU: We have provided all parameters used for BBMap, metaSpades, and all software included within the ATLAS Snakemake workflow (e.g., MetaBAT, Maxbin, dRep, Prodigal, linclust, egnog-mapper, among others). Please, see the Methods section, L679 onwards. Additionally, to align with and support the FAIR principles for scientific data management, the ATLAS configuration file containing all software parameters and the data sets and products generated from the raw sequence data have been now deposited at the INRAE institutional data repository powered by Dataverse with the following DOI: www.doi.org/10.15454/NGBSPC. In particular, to visualize the software parameters settings, see the files entitled “ATLAS_config.yalm”, and “ATLAS_dag.pdf”

22. Line 583: Was BBmap used here? Were the settings These same as was used for removing horse DNA?

AU: Yes, BBmap was used here, but settings were different as was used for removing horse DNA. The parameters used the quantification of each MAG abundance were set to pairlen=100, minid=0.9, mdtag=t, xstag=fs, nmtag=t, sam=1.3, ambiguous=best, secondary=t, saa=f, maxsites=10). The sample-specific median coverage of each MAG was then computed using pileup within BBMap with default parameters. This information is added in the Methods section. Please see page 15, L721-723.

23. Line 665: Please see my comment for line 226

AU: As indicated above, the PERMANOVA model was adjusted by factors that may affect the microbiome, e.g., age, sex, and dietary macronutrient intake. We have updated the description in the Methods sections: “The PerMANOVA test (a non-parametric method of multivariate analysis of variance based on pairwise distances) was implemented using the pairwiseAdonis2() function from the pairwiseAdonis (v. 0.4) R package tests the global association between ecological or functional community structure and groups. The model was adjusted by factors that may affect the microbiome, namely age, sex, and dietary macronutrient intake.”

24. Line 736-746: Sampling time must also be included in the model, or tests should be performed for T0 and T1 separately.'

AU: As noted above, fecal-related data were obtained solely at T0. Therefore, we could not include sampling time in the model, nor perform tests for T0 and T1 separately. Nonetheless, the text has been rephrased to make this issue clear.

Reviewer #2 (Remarks to the Author):

1. Mach and collaborators present a functional survey of the microbiome of horse athletes. Through pathway and genome reconstruction and analyses, they suggest clustering of microbiome profiles based on the dominance of Lachnospiraceae or the presence of rare species. They suggest the presence of rare species associates with improved cardiovascular capacity. The manuscript is well written, and the data great number of data analyses presented are robust for the most part. However, some info appears to be rather strong and speculative, such as the contentions of rare taxa contributing to enhanced athletic performance. In this regard, there is also lack of info on the nature of these rare taxa and the methods used to validate them or discard their origin due to possible contaminating sources, as there is indication of the use of negative controls.

AU: First, we thank you for your careful and thorough reading of this manuscript and the thoughtful comments and constructive suggestions. We deeply value your appreciation of the methodology and data analysis.

We agree that the first version of the manuscript was unkempt and needed some improvement and clarity, especially concerning the nature of the rare taxa and the methods to validate or discard their origin due to possible contamination. We have devoted more time removing speculative information that may distract the reader or be too extrapolative. Importantly, we have provided supplementary data and information to discard possible contamination sources in our data. Please, see below for further details.

Specific comments are as follows:

2. Title is speculative, the word “ensures” is rather strong, as the data mainly show associations

AU: We agree with this; hence the title has been changed to “Mining the equine gut metagenome: poorly-characterized taxa associated with cardiovascular fitness in endurance athletes”.

3. Abstract. L33: what does the article “they” refer to? The rare species? Also, can authors include info on what they consider rare (% threshold) and some examples of these rare taxa?

AU: Yes, “They” referred to the “rare species”. The reviewer’s concern evidenced that the term “rare” was inappropriate in our manuscript. All analyses were based on the dominant phylotypes. Dominant phylotypes included the top 25% of most highly abundant genera found in more than half of the metagenomic samples. Those phylotypes accounted for ~95% of the total classified sequences on average. Therefore, minor/rare microorganisms were discarded, reinforcing the notion that false-positive species predictions due to misalignment and contamination were likely not included in the analysis. In this regard, the authors have replaced the term “rare” with “poorly described/undescribed” taxa in horses throughout the manuscript, given that microbes occurring at low densities were discarded and treated as analytical annoyances. Furthermore, the Results and Methods sections have been updated to clarify these points. Please, see pages 5 (L201-205) and 16 (L793-801).

As the reviewer requested, we have described some examples of these yet undescribed taxa in horses. For example, see the text: “Then, we retained the most dominant microbial phylotypes. These phylotypes harbored 1,146 unique genera (accounting for 95% of the total classified sequences) with abundance and prevalence in the top 25% and 50% quantiles, respectively (Supplementary Table 9). They were represented mainly by prokaryotes (91%), a handful of eukaryotes (7.15%), and archaea (1.75%). In agreement with 16S rRNA gene sequencing data from the same samples, *Prevotella*, *Fibrobacter*, *Bacteroides*, and *Treponema* were the most prevalent genera in most individuals (Fig 3a and Supplementary Table 10 and 11). However, all individuals harbored large amounts of uncharacterized taxa in the horse, such as *Mediterraneibacter*, *Coprobacillus*, *Mucilaginibacter*, *Chitinophaga*, *Flavobacterium*, and *Enterocloster* (Fig 3a)” and the following text: “Consistent with the univariate analysis, it included the facultative bacterial predator *Lysobacter* and *Akkermansia*, generally regarded as health-promoting bacteria in athletes^{14–18}, along with anaerobic fungi (*Ophiocordyceps*, *Pseudogymnoascus*, *Trichoderma*, *Talaromyces*, *Rhodotorula*, *Exophiala*, *Puccinia*), methanogens (*Methanothermobacter*, *Methanothrix*) and algae (*Emiliania* and *Porphyra*).”.

4. -L34: Can the authors briefly elaborate on what specific “mitochondria-mediated mechanisms” they refer to?

AU: Yes, we can elaborate on specific mitochondrial-mediated mechanisms. In particular, we have re-elaborated the abstract while describing in more detail the mitochondria-mediated mechanisms. Please, see the new text: “Integration of microbial and host omic datasets suggested that functionally redundant microbiomes, predominantly enriched in

Lachnospiraceae taxa, were negatively associated with cardiovascular capacity. Conversely, more complex and functionally diverse microbiomes likely increased substrate availability and mitochondrial activity in different tissues, including the heart. In line with this hypothesis, more fit athletes showed increased expression of mitochondrial-related genes involved in energy metabolism, biogenesis, and Ca²⁺ cytosolic transport, all of which are necessary to improve aerobic work power, spare glycogen usage, and increase cardiovascular capacity. The results identify an associative link between horse endurance performance and its microbiome composition and gene function, laying the basis for nutritional interventions that could benefit endurance athletes”.

5. -It is not clear if what associated to improved endurance is the presence of rare species or the functions encoded by them.

AU: Unfortunately, we do not have experimental shreds of evidence to clarify if what is associated with improved endurance is the presence of specific species or the functions encoded by them. This caveat has been discussed in the main text: “While shotgun sequencing revealed a vast landscape of microbial diversity and functional genes in horse athletes, an improved reference genome database now needs to be built to refine the links between microbial strains and their functionalities”.

While unfeasible in elite athlete horses, we should test in animal models (i.e., mice following a specific exercise challenge) whether the inclusion or removal of particular taxa reduces the levels of metabolites in the FAO, OXPHOS, or citric acid cycle pathways and thus, the overall mitochondrial activity and expression. Nonetheless, we esteem that our associations predicted *in silico* open the way for mechanistic studies that will lead to a better understanding of the orchestrated molecular pathways that underpin endurance adaptations and contribute to holobiont biology.

6. Intro: L45: “160 km at 20 km/h” is not clear, do they mean 160km/h?

AU: We have rephrased the sentence: “Arabian horses have gained built-in biological mechanisms to compete at distances of up to 160 km in a single day, an effort comparable to a human marathon or ultra-marathon runners”.

7. L57-60: can authors elaborate a bit more on what microbially-produced metabolites are associated to those desirable phenotypic traits? Same for L77-78.

AU: Yes, we can elaborate on what microbially-produced metabolites are associated with those desirable phenotypic traits. Precisely, we have re-worked the paragraph: “Such gut microbiome-derived metabolites include short-chain fatty acids, dimethyl sulfone, trimethylamine oxide, indoles, tryptamine, oligosaccharides, peptidoglycans, and secondary bile acids^{19,20}, all of which affect host health²¹”.

Additionally, we have improved the L77-88 like this: “The interdependence of exercise performance and gut microbiota in horses is underscored by several lines of evidence^{3,11,22–25}, although the range and extent of this interplay are largely unknown. Recent findings suggest that gut microbial metabolites in endurance horses, e.g., acetate, valerate, dimethyl sulfone, trimethylamine oxide, formate, and secondary bile acids coupled with circulating free fatty

acids regulate mitochondrial function by preventing hypoglycemia ²³, which is the limiting factor for fatigue onset and, thus, athletic performance”.

8. Results: L103-104: Could you please clarify what this statement means? Do you mean that not all horses harbored the ~25M gene catalog, meaning that these are not core genes and therefore were not found across all horses? So is the gene catalog presented not really a “core” gene set?

AU: We fully agree that this point needed to be clarified. Our gene catalog contains ~25 M genes, and it does not represent a “core” set of genes. Less than 2 M of genes represented the core set of genes. In fact, individual horses harbored around half of the catalog genes ($n = 11,809,713$ genes), but the core genes present in all individuals consisted of less than 7.2% of the overall microbial gene pool ($n = 1,805,693$). Yet, these core genes showed highly conserved abundance rank structure across individuals, representing between 29.7 to 38.4% of the overall microbiome abundance. New text has been added to the main manuscript to clarify these data (see page 3; L117-126). Additionally, Fig 1 has been modified to depict the core genes and the genes that are unique to each individual.

Furthermore, your concern prompted the authors to contact Ang et collaborators ²⁶ to get the fasta file with the sequences of the microbial genes found in their cohort of feral and domesticated horses from different geographic locations. We used mmseqs2 (v.13.45111) to find genes at a 95% similarity threshold and 80% overlap between the two catalogs. The Methods section describes the parameters used (please see page 16; L766-770). Only 22.5% ($n = 922,362$) of the Ang et colleagues horse gut microbiome gene catalog overlapped with our gene catalog, indicating that the functional potential of gut microbiomes in horses is enormously vast and currently under-sampled (please, see page 3, L124-126)

Figure 1 – Taxonomic description of an expanded catalog of gut microbial genes: core genes and taxonomic annotation

(a) Contribution of different sample sources to gene content of the horse gut microbial gene catalog. Vertical blue bars represent the number of genes present in only one sample or shared between pairs of samples. Horizontal orange bars in the lower panel indicate the total number of genes contained in each sample; (b) Flower plot showing the number of core genes shared between all samples and those specific for each individual; (c) Visualization of the taxonomic assignment of Illumina trimmed paired reads in a Krona plot using the software tool Kaiju; (d) Lollipop plot showing the gene counts identified by the Kaiju resolved at the phylum level. Dots are colored by kingdom; (e) Treemap showing the taxonomic ranking of the main taxa in the gene catalog using Kaiju. The size of each box is proportional to the number of genes assigned to each taxon. The placement of boxes is arbitrary concerning boxes within the same taxonomic rank and does not correspond to any form of phylogeny or relatedness.

9. L107: Please state the database used. Also, if 61% were exact matches, does that mean 39% of reads were unknown? % of unknown/un-annotated reads should be described.

AU: We used the NCBI *nr_euk* reference database. Yes, 39% of the gene sequences were unclassified. This information has been added to the text: “Taxonomic annotation of the microbial gene catalog was implemented with Kaiju²⁷ using a greedy mode and the NCBI *nr_euk* reference database. While 39% of the gene sequences were unclassified, this approach annotated 61% of the gene sequences (Fig 1c) and revealed a diverse community of 95 phyla encompassing 1,110 families and 4,179 unique genera (Supplementary Table 3). Strikingly, > 90% of these genera have not been previously described in horses. Among classified genes, bacteria (95.58% of genes, $n = 2,927$ genera) defined most of the assemblage in terms of

abundance and diversity, followed by a handful of eukaryotes (2.55% of genes, $n = 1,081$ genera), and archaea (1.20% of genes, $n = 170$ genera).”

10. L140: why are authors reporting pathways associated with endocrine system and neurodegenerative diseases? These are eukaryote pathways that show up because of shared prokaryote and eukaryote (human) genes involved in diverse pathways- these pathways, as inferred from bacterial metagenomes should be filtered out.

AU: We thank the reviewer for bringing this mistake to our attention. These pathways have been filtered out from Supplementary Table 5, and the text has been modified accordingly. We have clarified this in the text and rephrased the sentence as follows:

“Most KOs had essential microbial gut functions, including metabolism (transport and metabolism of amino acids, carbohydrates, nucleotides, and lipids), genetic information processing, and signaling (Supplementary Table 5). We also identified potential functions related to the biosynthesis of secondary metabolites, quorum sensing, and prokaryotic defense system, which help maintain the microbiome's structure and the host's health (Fig 2a)”.

11. L156-158 (Figure 1d), suggestion: Perhaps it maybe be more interesting, in the context of the question pursued by authors, to show the CAzyme data (FS2) as main figure, 1d for instance? AMR genes analyses are interesting but unclear what these data brings to understand the effect of the microbiome on athlete performance (appears to be rather distracting since no reference on this is made in the rest of the manuscript). AMR genes are expected to ubiquitous in all microbiomes, regardless of exposure, so it is not clear what the intent is with this analysis.

AU: The reviewer is correct. Therefore, we have moved the CAzyme and KOs plots in the main text. Please see the new Fig 2.

We agree that AMR genes are expected to be ubiquitous in all microbiomes. However, we detected an extended-spectrum of β -lactamase (ESBL) *bla*_{ACI-1} found in several Negativicutes (Gram-negative Firmicutes) but rarely detected in animal or human gut microbiomes. Effectively, AMR genes do not expand our understanding of the microbiome's effect on athlete performance. However, since the first section of the paper aims to describe the horse gut microbiome gene catalog and MAG repertoire irrespective of athletic performance, we believe that the AMR description can be kept. However, we can remove it to avoid misleading the reader if the reviewer considers so.

Figure 2 – Functional diversity of an expanded catalog of gut microbial genes, antimicrobial genes, and MAGs landscape
 (a) Frequency of KEGGs pathways in the gut microbial gene catalog estimated from the orthologous groups (KOs). The horizontal bars represent the absolute number of genes found in each of the KEGG pathways; (b) Frequency of total carbohydrate-active enzyme (CAZymes) families estimated in the gut microbial gene catalog. CAZymes are colored according to their class; (c) Heatmap shows each individual's normalized counts of antimicrobial resistance (AMR) genes based on the ResFinder database. The left column depicts the AMR class; (d) Phylogenetic tree of the 372 MAGs detected. The outer cycle boxplots represent the abundance of each MAG. Boxplots are colored according to phyla; (e) Sankey diagram showing the numbers of MAGs with completeness between 50-90% or > 90%. Only a relatively small fraction of the MAGs (73 out of 372) was annotated at the genus level; (f) Circular stacked bar plot of the MAGs phyla abundance detected for each individual in the cohort.

12. L199-200: Can authors provide brief description of those macronutrient intake analyses? Up to this point it was not clear these analyses had been made. A brief intro (few lines) stating which analyses, why and how they were conducted ? Also, it is clear

that macronutrient intake did not differ between horses in the low and high performance clusters, but why wasn't that factor included in the PERMANOVA analyses validating the microbial clusters? That analysis probably should be included to test for the same clusters, based on CAZymes and KOs.

AU: Following the reviewer's comment, we have added a new paragraph in the results section describing in more detail the elite athletes' diet on training and non-training days: "To tie together the basal dominant gut phylotypes with the horse performances, we investigated whether diversity, compositional and functional differences between clusters were driven by nutrient intake or any horse-centered parameter. Recorded dietary data showed that participants had a constant standard varied macronutrient diet on training and non-training days [mean \pm SD (% dry matter (DM) basis): 53.7 \pm 19.3 MJ/day, 939 \pm 42 g/day of protein (11 \pm 1.9%), 678 \pm 20 g/day of fat (8 \pm 1.6%), 2,190 \pm 706 g/day of hydrolyzable carbohydrates (26 \pm 8.9%), and 2,320 \pm 877 g of fermentable fiber (27 \pm 2.8%)]. Individuals were fed 8.43 \pm 2.99 kg/day, with an 80:20 forage to concentrate ratio (Supplementary Table 1)".

Yes, macronutrient intakes were included in the PERMANOVA analyses to validate the microbial clusters, together with the sex and age. As described in reviewer #1's comment #11, the model included cardiovascular fitness (composite.heart.binaire) adjusted by the effect of sex, age of the athlete, energy intake (UFC), and each macronutrient dietary intake (please see below). The breed was not included in the model since values were unique for half-Arabian and Shagya-Arabian. In sum, the model states that gut microbiome structure is mainly independent of age, sex, and diet. The same model was tested for CAZymes and KOs. The results are reported as follows: "None of the macronutrient intakes were statistically different between the two clusters ($p > 0.05$, two-sided Wilcoxon rank-sum test), and none of them were the major driving forces of the microbiome composition (PerMANOVA; $R^2 = 0.102 \pm 0.089$ and p -values ranging from 0.055 to 0.82; Supplementary Fig 5a-d). Likewise, KOs (PerMANOVA; $R^2 = 0.034 \pm 0.010$ and p -values ranging from 0.940 to 0.969) and CAZymes profiles (PerMANOVA; $R^2 = 0.030 \pm 0.016$ and p -values ranging from 0.250 to 0.971) were also independent of nutrient intakes. Gut community composition was neither linked to horse sex (PerMANOVA; $R^2 = 0.2486$, $p = 0.266$), or age of the athletes (PerMANOVA; $R^2 = 0.0952$, p -value = 0.409; Supplementary Fig 5e-f)".

This comment encouraged us to create Supplementary Fig S5, which shows no effect of daily nutrient consumption on microbiome composition and function.

13. L229: Shouldn't it read "higher abundance OF"?

AU: The sentence has been re-phrased in this way: "In contrast to the overwhelming diversity observed in cluster 1, a dwarfed biodiversity defined cluster 2 (Supplementary Table 12 and Supplementary Fig 3a) that assembled around highly efficient fiber degraders (members from the *Lachnospiraceae* taxa: *Anaerostipes*, *Butyrivibrio*, *Blautia*, *Coprococcus*, *Dorea*, *Eubacterium*, *Hespellia*, *Lachnospira*, *Oribacterium*, *Roseburia*, and *L-Ruminococcus*) and downstream users of degradation products (*Prevotella*, and *Treponema*)¹³ (DESeq2, adjusted $p < 0.05$; Supplementary Table 12 and Supplementary Fig 3c)".

Discussion

14. L336: What is meant by “tissue repair” ? what microbial functions reflect that end?

AU: Tissue repair was meant to “repair” the increased intestinal permeability commonly found in endurance athletes²⁸. An excessive release of stress hormones induced by physical and psychological stress during endurance exercise often cause lipopolysaccharides to translocate outside of the gastrointestinal tract, triggering immune and inflammatory responses that result in tight junction loss and increased intestinal permeability. Anyway, “tissue repair” has been deleted from the text. The new text is: “This catalog captured a wide array of functions associated with complex carbohydrate fermentation and functional capacities that enabled more energy extraction from dietary, microbial, and host resources, which likely provided a performance advantage for athletes.”

However, your comment made us reflect on whether some microbial functions could reflect that end. It is worth noting that coupled with the enhanced metabolic plasticity, cluster 1 individuals presented a significantly lower abundance of horse DNA in the shotgun raw sequencing data ($p = 0.05$, two-sided Wilcoxon rank-sum test). These results likely evoke a lower amount of epithelium disruption or intestinal inflammation, commonly observed in endurance athletes. So, no microbial functions reflect that end but the amount of host DNA. As described in the discussion, given the small sample size without additional information, it is difficult to conclude the reason for this finding. These results are depicted on page 6 (L254), and in Supplementary Fig 4c.

15. L357-363: This contention is a bit vague. Although it may seem intuitive as per the results and what authors want to convey, can they elaborate bit more on what these interventions may look like? How will these interventions will “create more space” for rare taxa? What specific probiotics/fibers are they referring to that may expand rare taxa while constraining Lachnospiraceae? How would fecal transplants increase rare taxa and inhibit dominant Lachno? This is may be a vague statement since; 1) Lachnospiraceae are dominate phylotypes in most herbivore mammals, and especially equines 2) authors have not really elaborated or expanded on the true abundance, nature and function of these rare taxa, 3) The number of horses that are deemed as high performers and that contain these rare taxa is only three, 4) it is impossible to validate the role of these taxa based on the 16S rRNA dataset (what rare taxa could be validated in these dataset?)

AU: The reviewer is correct. Therefore, we have elaborated a bit more on how these interventions may look like: “in light of these findings, nutritional interventions to reduce *Lachnospiraceae* taxa like *Dorea* and *L-Ruminococcus* while creating more space for non-core species will likely be required to increase microbiome diversity, functional plasticity, and athletic performance. Diet changes can promote swift changes in *Lachnospiraceae* abundance¹: consumption of dietary pectins could strongly reduce the fecal abundance of *Ruminococcus*, *Blautia*, and *Roseburia* (all pertaining to the *Lachnospiraceae* family)²⁹. Conceptually, the partial replacement of diet-derived polysaccharides needed to expand *Lachnospiraceae* taxa (e.g., starch, inulin, xylans, and arabinoxylan) by other structurally diverse dietary fibers (e.g., soluble pectins such galactan, arabinan, and arabinogalactan) could be the front-line for nutritional interventions. Owing to the minor involvement of the *Lachnospiraceae* phylotypes

in proteolytic metabolism³⁰, increasing protein consumption could be another way to optimize the microbiome function in athletes”.

The use of fecal transplants to increase rare taxa and inhibit dominant *Lachnospiraceae* has been removed from the text, as we have no evidence of their feasibility.

Following your suggestions, we have expanded the information on the true abundance of all these poorly described microorganisms that were highly represented in the more fit individuals, including interquartile range variation, median, mean, SD, min, and max (please see Supplementary Table 12). Nevertheless, we could not associate the actual abundance of these taxa with their function from our data. The metabolites derived from such species or produced by the host and biochemically modified by them remain unknown. While fecal metabolome could provide a functional readout of microbial activity and be used as an intermediate phenotype mediating host-microbiome interactions, still, it does not identify the natural-microbiome-enzymatic process associated with each microorganism. To further study these mechanisms, the germ-free model might again be helpful. One could monocolonise mice with one or different bacteria and then look at the metabolome and running performance. One could also precisely dissect the contribution of enzymes to any observed effect.

Yes, horses that were deemed as high performers and contained these rare taxa were only three, and it was impossible to validate the role of these taxa based on the 16S rRNA dataset. The “rare” taxa that could be confirmed in the validation set were *Mogibacterium*, the yet undefined members of *Rhodospirillaceae*, *Enterobacteriaceae*, *Planococcaceae*, and *Sphingobacteriaceae* families. This information has been modified in the new text: “This analysis only supported the presence of *Mogibacterium* and the yet undefined members of *Rhodospirillaceae*, *Enterobacteriaceae*, *Planococcaceae*, and *Sphingobacteriaceae* families in more fit individuals but not the other 252 poorly-described bacteria members. This likely reflects an under-representation of these taxa in existing 16S rRNA reference databases. Conversely, the SPLS-DA corroborated the enrichment of Clostridiales, *Erysipelotrichaceae*, and *Lachnospiraceae* taxa in less competitive individuals. Markedly, genera such as *Blautia*, *Butyrivibrio*, *Coprococcus*, *Dorea*, *Desulfovibrio*, *Hespellia*, *Lachnospira*, and *L-Ruminococcus* (all pertaining to the *Lachnospiraceae* family) were consistently depleted in less fit athletes in both the discovery and the validation sets. Additionally, *Dorea* and the polyphyletic *Ruminococcus* were consistent hallmark genera of less competitive individuals (DESeq2, adjusted $p = 0.0491$; Supplementary Fig 7e-f) across both discovery and validation cohorts”.

16. In this regard, both in the results and discussion sections there should be some lines devoted to describe discuss the abundance (ranges, averages) of these rare taxa and info on the environments they have been normally detected in. There should also be some discussion as to why authors discard the possibility of these taxa being the result of contamination? Why didn't the authors use negative (blank extraction) controls? Addressing these issues is key as blooms in microbial contamination (e.g. kit associated contaminants) can significantly bias alpha diversity, which authors claim to be a marker of high endurance/athletic performance.

AU: We have devoted some lines to describing these taxa's abundance, and we have created the Supplementary Fig 3a-b. Additionally, the Supplementary Table 12 has been extended by adding columns corresponding to the mean, SD, IQ1, median, IQ3, min, and max for each differentially expressed taxa, including the $n = 318$ poorly described ones. Unfortunately, we have not been able to tell the environments in which they have been normally detected as information is scant, and we are already exceeding the recommended length of the manuscript.

Figure S3 - Interconnection between cardiovascular fitness, microbiome composition, and functionality

(a-b) Line plots showing the relationship between frequency (y-axis) and the relative abundance (x-axis) of the 318 upregulated genera in cluster 1 compared to cluster 2, averaged across cluster 1 (a) and cluster 2 (b); (c) Dot plot representation of log-transformed fold change of dominant phylotypes significantly differed between the individuals with reduced and improved cardiovascular capacity. The logs of fold changes lying above 0 indicate that phylotypes were more abundant in individuals with reduced cardiovascular capacity than those with improved cardiovascular fitness. By contrast, the negative logs of fold changes lying between indicate that the phylotypes abundance was lower in individuals with reduced cardiovascular capacity than more fit participants. Phyla color dots; (d) Dot plot representation of log-transformed fold change of CAZymes significantly differed between individuals with reduced and improved cardiovascular capacity. The logs of fold changes lying between 0 and 1.5 indicate that CAZymes were more abundant in less fit individuals. By contrast, the logs of fold changes between 0 and -1.5 show that the CAZymes abundance was lower in individuals with reduced cardiovascular capacity than in more fit individuals. Classes color dots.

Significantly, we have added arguments and data to justify that these taxa are not the result of contamination. First, we selected the most dominant microbial phylotypes, which circumvented the low methodological-based signal-to-noise ratio and ensured a robust analysis of the microbial communities. Dominant phylotypes included the top 25% of most highly abundant genera found in more than half of the metagenomic samples. Please see the text: “We

first profiled microbial taxa in each metagenomic baseline sample at the taxonomic (NCBI *nr_euk* reference database) and functional (EggNOG database). Then, we retained the most dominant microbial phylotypes. These phylotypes harbored 1,146 unique genera (accounting for 95% of the total classified sequences) with abundance and prevalence in the top 25% and 50% quantiles, respectively (Supplementary Table 9).”.

Second, we ran the decontam R package on the 16S rRNA data in which negative controls were available. Only 6 ASV were statistically classified as contaminants (see Supplementary Fig 9), although their frequency plots showed they were non-contaminants. All this information and results have been added to the main manuscript (see page 13, L 643-666). Given that shotgun and 16S rRNA sequencing were based on the same DNA extraction, we can discard blooms in microbial contamination before sequencing (*e.g.*, kit-associated contaminations) and overestimated alpha-diversity. Regrettably, we could not use negative (blank extraction) controls for whole-genome sequencing due to financial constraints.

Figure S9 – Identification of 16S rRNA contamination data

The distribution of the frequency of each ASV was calculated as a function of the input DNA concentration using the decontam R package. In this plot, the dashed black line shows the model of a noncontaminant sequence feature for which frequency is expected to be independent of the input DNA concentration. The red line shows the model of a contaminant sequence feature, for which frequency is expected to be inversely proportional to input DNA concentration, as contaminating DNA will make up a more significant fraction of the total DNA in samples with very little total DNA.

17. L367-368: What microbial pathways and genes are involved explicitly in mitochondrial biogenesis and energy resilience? This is not clear in the analyses nor supported by existing data.

AU: We agree with the reviewer that this information was not evident in the text. These results came from the unfiltered eukaryote KOs table. Given that the author brought this mistake to our attention, these pathways have been removed from Supplementary Table 5 and the manuscript.

18. L373-380: this contention is highly speculative, authors have provided no direct evidence that these are taxa have any capacity to break down host glycans or that the metabolism of such substrates can significantly contribute to the energetic landscape of the host, or to mitochondrial function. If anything one would expect that energy harvest from dietary glycans far exceed any contributions (if any) from metabolizing animal glycans. Also, what metabolic products would be released by these metabolic route, that are not already released by feed fermentation?

AU: We have edited the text following the reviewer's comments (please see page 5, L237-253). Effectively, we cannot provide direct evidence that these are taxa that can break down host glycans and affect mitochondrial function. However, cluster 1 had significantly higher amounts of encoding CAZymes for host glycans degradation (14.15%) than members of cluster 2 (9.63%), as well as KOs related to glycan biosynthesis and metabolism (K09953, K18770, K12309, K12551, K01137, K03276, K12985, K14459). Cluster 1 also exhibited higher enrichment of microbial pathways related to amino acids metabolism (K12256, K15226, K03852, K08688, K18049, K21062), energy production, e.g., genes within lipid metabolism (K10781, K16795, K01795, K01049, K12309), propanoate metabolism (K11264, K01659, K00382, K01720), fructose and mannose (K19633) and pyruvate (K02594), as well as and methane production (K00202, K00204, K00583, K05884). Therefore, we believe that individuals from cluster 1 enjoyed a broader availability of resource compounds, including SCFAs, host glycans, dietary and microbial proteins, lipids, propanoate, pyruvate, and hydrogen, to produce higher relevant goods production for host energy requirements.

19. Many of the rare taxa found are involved in methane metabolism. Can authors elaborate on possible contributions of these metabolic pathway, from an energetic stand point, on horse endurance? Especially in ten context of lack of differences in SCFA profiles.

AU: We thank the reviewer for raising this interesting point. We agree that the increase in methane-related taxa was mentioned in passing. For instance, methanogens belonging to the Methanomicrobiales (*Methanotherix*), Methanobacteriales (*Methanosphaera*, *Methanobacterium*, *Methanothermobacter*), and Thermococcales (*Thermococcus*) orders were increased in the cluster 1. Counterintuitively, the enrichment of methane-producing taxa was not associated with differences in SCFA profiles. Still, it was accompanied by a higher abundance of two hydrogen providers, known to interact with methanogens³¹, the ciliated protozoa, and *Mogibacterium* (also found in our validation cohort). It remains to be determined how these inter-kingdom interactions affect energy harvest efficiency in athletes. Beyond

methane's effects on energy metabolism, several shreds of evidence suggest that methane exerts anti-inflammatory, anti-apoptotic, and anti-oxidative effects in the gut³². Therefore, methane production could decrease gut inflammation resulting in less intestinal permeability and fecal shedding of leukocytes in athletes, conferring protection from the inflammatory effects of endurance exercise²⁸—this hypothesis matched with the lower abundance of horse DNA in feces. Given the small sample size without additional information, it is difficult to conclude the reason for this finding. The new text is added in the results and discussion sections (please see page 10, L453-460).

20. L402-406: Does the finding of rare taxa correlates with the remarkable depth of this metagenomic dataset ? In other words, would these rare taxa be detected in any horse population regardless of phenotype?

AU: We believe that the finding of poorly defined taxa is related to the remarkable depth of this metagenomic dataset. To answer the reviewer's question, we first check the correlation coefficient between the number of microbial genes and the sequencing depth (Gb). The supplementary Fig 1c shows that the higher the sequencing depth, the higher the number of microbial genes ($r^2 = 0.38$, $p < 0.05$). We then contacted Ang et collaborators²⁶ to share their fasta file containing the microbial genes found in their feral and domestic horses cohort. They sent us a fasta file containing 4,086,123 genes. On average, they had 11.06 GB of raw sequences per sample. We have more than twice the raw data per sample, and in fact, the average number of genes per individual was almost three-fold (~11M).

Only 22.5% ($n = 922,362$) of their genes overlapped with our gene catalog at a 95% similarity threshold and 80% overlap. In other words, we believe that a high percentage of these taxa will not be detected in any horse population unless high metagenomic depths are applied. We believe that the number of microbial genes, poorly-described taxa, and MAGs will likely increase as more samples are analyzed, similar to what has been found for the other livestock gut metagenomes⁶⁻⁹. All these results have been included in the new version of the manuscript and discussed afterward.

21. In this regard, it is not clear how the discriminatory power of rare taxa was validated. So far it seems the prevalence of rare taxa in high performers was assessed via loadings in the MDS projection, but there doesn't seem to be a procedure or test (e.g. via DESeq, ANCOM, LEfSe) showing how discriminating taxa were validated. It is assumed that figure S8 accomplish this goal, but such figure does not appear to be referenced in the text (apologies if I am missing this)

AU: As the reviewer suggested, coupled with the sPLS-DA, we applied DESeq2 to the data. Unfortunately, we could not find any taxa statistically significant in high performers, as assessed through DESeq2. In contrast, *Dorea* and the polyphyletic *Ruminococcus* were consistent hallmark genera of less competitive individuals (DESeq2, adjusted $p = 0.0491$) across both discovery and validation cohorts. Please, see the new text on page 7 and Supplementary Fig 7e-f.

Figure S7 – Validation set with 22 independent elite athletes: effect of cardiovascular fitness on the gut microbiome based on 16S rRNA amplicon sequencing

(a) Boxplot representation of the composite of cardiovascular capacity in the validation set: “L” corresponds to “low,” M” to “Medium,” and “H” to “high.” The composite of the cardiovascular capacity was made by combining the post-exercise heart rate, the cardiac recovery time, and the average speed during the race. Adjusted p values from two-sided Wilcoxon rank-sum test; (b) PCoA ordination analysis (Bray-Curtis distance) of the relative abundances estimated with the 16S rRNA sequencing in the validation set. Points denote individual samples, which are colored according to the clustering group: H (red), M (green), and L (blue); (c) Bray-Curtis distance to the centroid of the gut microbial ASVs estimated by the 16S rRNA gene between the Low, Medium and High groups. Boxes show median and interquartile range, and whiskers indicate 5th to 95th percentile. Adjusted p values from Tukey’s Honest significant differences tests; (d) Multivariate sparse partial least-squares discriminant analysis (sPLS-DA) loading plot showing the contributing genera towards the separation between individuals with better cardiovascular fitness (red color) and less fit individuals (blue color). Bar length indicates loading coefficient weight of selected genus, ranked by importance, bottom to top; (e-f) Boxplots of *Ruminococcus* phylotypes and *Dorea* abundance, which were significantly different between individuals with better cardiovascular fitness (red color) and less fit individuals (blue color) at the basal time (DESeq2, adjusted p -values < 0.05). Boxes show median and interquartile range, and whiskers indicate 5th to 95th percentile. The box color indicates the group: H (red), M (green), and L (blue)

22. L411: There are conflicting views regarding the potentially beneficial role of Akkermansia, as its abundance has also been associated with the erosion of the mucus layer due to lack of fermentable fiber:

AU: We agree with the reviewer. We have modified the results section stating: “*Akkermansia*, generally regarded as health-promoting bacteria in athletes^{14–18}”, but we have removed the *Akkermansia* paragraph from the discussion to avoid conflicting views.

23. L414: How can an “intimate” connection between gut microbes and mitochondrial function., at the systemic level, be assessed? Is that possible at all? This contention is highly speculative.

AU: Because of the reviewer’s concerns, we have toned down the discussion. We believe that microbial metabolites can be absorbed and may increase the bioavailability of mitochondrial resources in any kind of tissue, including the heart. Several studies have uncovered biologically important links between skeletal muscle (including the heart) and the gut microbiota, suggesting the gut microbiota respond to an exercise challenge and have reciprocal roles in fuel availability, muscle function, and, therefore, endurance performance^{33–37}.

24. Authors are encouraged to include a study limitation section based on the concerns raised above – including, among others, validation of rare taxa as markers of high performers, inability to assess their true function or provenance, their true (or likely) impact on the systemic landscape of the host and the fact that rare taxa are only observed in three horses.

AU: Following your suggestion, we discussed our study caveats throughout the discussion. For example: “Owing to the difficulty of sampling elite horses (especially after long races), these observations were made on a small sample size, on a single occasion with a few samples for independent validation. Additional continuous long-term measurements will hence be needed to provide strong support to this novel aspect of horse physiology.” and “Because similar results may arise from DNA contamination before or during sequencing, we restricted the analysis to the cohort’s most prevalent and dominant phylotypes”. Another limit highlighted in the text is that one: “Importantly, even if the gastrointestinal tracts in horses and humans are composed of anatomically similar organs, there are some prominent differences. In contrast to what happens in humans, the cecum is large relative to the total gastrointestinal tract in horses; therefore, confounding factors should be considered for further relevance and translation of human dietary applications.”. Lastly, we have stated that: “While shotgun sequencing opened the can of microbial diversity and functional genes in horse athletes, an improved reference genome database now needs to be built to refine the links between microbial strains and their functionalities.”.

Methods

25. L554: Info on negative controls for the microbiome procedure should be included

AU: The information is now included in the Methods section. Please see page 13.

We have stated: “A negative control sample alongside biological samples at the DNA extraction and PCR steps was considered in attempts to control DNA contamination before and after sequencing. Additionally, contamination was minimized through laboratory techniques such as UV irradiation of material, ultrapure water, the DNA-free Taq DNA polymerase, and the separation of pre-and post-PCR areas”. Additionally, we have described: “The negative control sample did not yield a band on the agarose gel, and the concentration of the purified amplicon was undetectable (< 1 ng/μL). Nevertheless, the decontam (v. 1.14.0) R package was used to identify and visualize possible contaminating DNA features in the negative control sample. The function isContaminat() was used to determine the distribution of the frequency of each contaminant feature as a function of the input DNA concentration. Only 6 ASV were

statistically classified ($p < 0.05$) as contaminants, although their frequency plots showed they were non-contaminants (Supplementary Fig 9).”

26. L670: Where is this interindividual/PAM clustering analyses included and what is its purpose?

AU: We have deleted this interindividual/PAM clustering analysis. It was initially included to reinforce the robustness of the two-community clusters further.

Reviewer #3 (Remarks to the Author):

1. Using elite racehorses as a model, this work sought to build the first gene microbial catalog for this animal in an effort to better elucidate cardiovascular fitness. In line with Communications Biology, this manuscript represents a significant advance in the integration of systems biology and exercise science, presenting novel findings. While quite limited in sample size, the considerations and new evidence this work presents have important implications for future microbiome research in relation to exercise and physical activity (in general). For instance, microbiomes that supported rarer species were able to provide an expanded repertoire of metabolic pathways and mitochondrial potential as compared to those less diverse and higher in abundance of taxa in the Lachnospiraceae family. While not conducted in humans, the use of an equine model is innovative and well suited for the present examination. Moreover, the comparative physiology aspects of this work are commendable and were well received by this reviewer.

AU: We thank you for recognizing our work.

2. Overall, the manuscript is clearly written and of appropriate length. However, in relation to not overselling the thesis of this work, extrapolating findings from this model needs more care. For example, while the introduction generally sets the stage well for the workflow, one critical issue that should be noted is the distinct differences in gastrointestinal structure and physiology of those species better equipped to endure high-end cardiovascular output. In the spirit of the comparative aspects of this work, clearly horse and human GI tracts have distinctions. Perhaps highlighting the important similarities of successful phenotypes could better improve such comparisons. Insertion could in the paragraph starting at line 70, where Arabian horses are proposed as the in vivo model. This could also be reinforced in the discussion. With these inclusions this work will have done a fair job in the treatment of the previous literature.

AU: Given the reviewer’s suggestion, we have modified the introduction while highlighting the essential similarities of successful phenotypes. Please, see the new text: “Through years of selective breeding for athletic performance and tailored training practices, Arabian horses have gained built-in biological mechanisms to compete at distances of up to 160 km in a single day, an effort comparable to a human marathon or ultra-marathon runners³⁸. Like humans³⁹, these equine athletes display well-adapted physiological abilities, including processing large volumes of oxygen required for aerobic metabolism and markedly larger hearts than those less physically active. Unique to horses is their quadrupedal nature of locomotion, the coupling of respiration to stride frequency, and the pronounced increase in pulmonary artery pressure.

These equine-specific features place different physiologic loads on the heart during endurance than those observed in humans³⁹.”

Additionally, in line with your concerns regarding distinct differences in gastrointestinal structure, we have added in the discussion the following text: “Importantly, even if the gastrointestinal tracts in horses and humans are composed of anatomically similar organs, there are some prominent differences. In contrast to what happens in humans, the cecum is large relative to the total gastrointestinal tract in horses; therefore, confounding factors should be considered for further relevance and translation of human dietary applications.”.

3. The methods are generally well described, transparent, and performed in a sound manner. For the purpose of reproducibility, it would be a plus to have the main analyses available on a platform such as GitHub or even the outputs on Qiita.

AU: Thank you for appreciating the methodology, analysis, and interpretation. Following the reviewer’s suggestion, data sets and products generated from the raw sequence data are now available at the INRAE institutional data repository powered by Dataverse with DOI: www.doi.org/10.15454/NGBSPC.

(<https://data.inrae.fr/dataset.xhtml?persistentId=doi:10.15454/NGBSPC>). They have been appropriately specified in the text where required. All other data is available in the Supplementary Data.

References used in the referees’ letter

1. Steelman, S. M., Chowdhary, B. P., Dowd, S., Suchodolski, J. & Janečka, J. E. Pyrosequencing of 16S rRNA genes in fecal samples reveals high diversity of hindgut microflora in horses and potential links to chronic laminitis. *BMC Vet. Res.* **8**, 231 (2012).
2. Venable, E. B. *et al.* Effects of Feeding Management on the Equine Cecal Microbiota. *J. Equine Vet. Sci.* **49**, 113–121 (2017).
3. Mach, N. *et al.* Priming for welfare: gut microbiota is associated with equitation conditions and behavior in horse athletes. *Sci. Rep.* **10**, 8311 (2020).
4. Kaoutari, A. El, Armougom, F., Gordon, J. I., Raoult, D. & Henrissat, B. The abundance and variety of carbohydrate-active enzymes in the human gut microbiota. *Nat. Rev. Microbiol.* **11**, 497–504 (2013).
5. Leitch, E. C. M. W., Walker, A. W., Duncan, S. H., Holtrop, G. & Flint, H. J. Selective colonization of insoluble substrates by human faecal bacteria. *Environ. Microbiol.* **9**, 667–679 (2007).
6. Xie, F. *et al.* An integrated gene catalog and over 10,000 metagenome-assembled genomes from the gastrointestinal microbiome of ruminants. *Microbiome* **9**, 1–20 (2021).
7. Li, J. *et al.* A catalog of microbial genes from the bovine rumen unveils a specialized and diverse biomass-degrading environment. *Gigascience* **9**, 1–15 (2020).
8. Chen, C. *et al.* Expanded catalog of microbial genes and metagenome-assembled genomes from the pig gut microbiome. *Nat. Commun.* **12**, 1–13 (2021).

9. Xiao, L. *et al.* A reference gene catalogue of the pig gut microbiome. *Nat. Microbiol.* **1**, 1–6 (2016).
10. Hawley, J. A., Lundby, C., Cotter, J. D. & Burke, L. M. Maximizing Cellular Adaptation to Endurance Exercise in Skeletal Muscle. *Cell Metab.* **27**, 962–976 (2018).
11. Janabi, A. H. D., Biddle, A. S., Klein, D. & McKeever, K. H. Exercise training-induced changes in the gut microbiota of Standardbred racehorses. *Comp. Exerc. Physiol.* **12**, 119–130 (2016).
12. Poulsen, M. *et al.* Methylotrophic methanogenic Thermoplasmata implicated in reduced methane emissions from bovine rumen. *Nat. Commun.* **4**, (2013).
13. Mizrahi, I., Wallace, R. J. & Morais, S. The rumen microbiome: balancing food security and environmental impacts. *Nat. Rev. Microbiol.* **19**, 553–566 (2021).
14. Clarke, S. F. *et al.* Exercise and associated dietary extremes impact on gut microbial diversity. *Gut* **63**, 1913–1920 (2014).
15. Bressa, C. *et al.* Differences in gut microbiota profile between women with active lifestyle and sedentary women. *PLoS One* **12**, 1–20 (2017).
16. Munukka, E. *et al.* Six-week endurance exercise alters gut metagenome that is not reflected in systemic metabolism in over-weight women. *Front. Microbiol.* **9**, 2323 (2018).
17. Petersen, L. M. *et al.* Community characteristics of the gut microbiomes of competitive cyclists. *Microbiome* **5**, 98 (2017).
18. Karl, J. P. *et al.* Changes in intestinal microbiota composition and metabolism coincide with increased intestinal permeability in young adults under prolonged physiological stress. *Am. J. Physiol. - Gastrointest. Liver Physiol.* **312**, G559–G571 (2017).
19. Chen, H. *et al.* A Forward Chemical Genetic Screen Reveals Gut Microbiota Metabolites That Modulate Host Physiology. *Cell* **177**, 1217–1231.e18 (2019).
20. Donia, M. S. & Fischbach, M. A. Small molecules from the human microbiota. *Science (80-.)*. **349**, 395 (2015).
21. Frampton, J., Murphy, K. G., Frost, G. & Chambers, E. S. Short-chain fatty acids as potential regulators of skeletal muscle metabolism and function. *Nat. Metab.* **2**, 840–848 (2020).
22. Plancade, S. *et al.* Unraveling the effects of the gut microbiota composition and function on horse endurance physiology. *Sci. Rep.* **9**, 9620 (2019).
23. Mach, N. *et al.* Understanding the Holobiont: Crosstalk Between Gut Microbiota and Mitochondria During Long Exercise in Horse. *Front. Mol. Biosci.* **8**, 656204 (2021).
24. Mach, N. *et al.* Gut microbiota resilience in horse athletes following holidays out to pasture. *Sci. Rep.* **11**, 5007 (2021).
25. Janabi, A. H. D., Biddle, A. S., Klein, D. J. & McKeever, K. H. The effects of acute strenuous exercise on the faecal microbiota in Standardbred racehorses. *Comp. Exerc. Physiol.* **13**, 13–24 (2017).
26. Ang, L. *et al.* Gut Microbiome Characteristics in feral and domesticated horses from different geographic locations. *Commun. Biol.* **5**, 1–10 (2022).
27. Menzel, P., Ng, K. L. & Krogh, A. Fast and sensitive taxonomic classification for metagenomics with Kaiju. *Nat. Commun.* **7**, 11257 (2016).
28. Clark, A. & Mach, N. Exercise-induced stress behavior, gut-microbiota-brain axis and

- diet: a systematic review for athletes. *J. Int. Soc. Sports Nutr.* **13**, 43 (2016).
29. Larsen, N. *et al.* Potential of pectins to beneficially modulate the gut microbiota depends on their structural properties. *Front. Microbiol.* **10**, 1–13 (2019).
 30. Vacca, M. *et al.* The controversial role of human gut lachnospiraceae. *Microorganisms* **8**, 1–25 (2020).
 31. Newbold, C. J., De la Fuente, G., Belanche, A., Ramos-Morales, E. & McEwan, N. R. The role of ciliate protozoa in the rumen. *Front. Microbiol.* **6**, 1313 (2015).
 32. Boros, M. *et al.* The anti-inflammatory effects of methane. *Crit. Care Med.* **40**, 1269–1278 (2012).
 33. Okamoto, T. *et al.* Microbiome potentiates endurance exercise through intestinal acetate production. *Am. J. Physiol. - Endocrinol. Metab.* **316**, E956–E966 (2019).
 34. Nay, K. *et al.* Gut bacteria are critical for optimal muscle function: A potential link with glucose homeostasis. *Am. J. Physiol. - Endocrinol. Metab.* **317**, E158–171 (2019).
 35. Barton, W. *et al.* The effects of sustained fitness improvement on the gut microbiome: A longitudinal, repeated measures case-study approach. *Transl. Sport. Med.* **4**, 174–192 (2021).
 36. Estaki, M. *et al.* Cardiorespiratory fitness as a predictor of intestinal microbial diversity and distinct metagenomic functions. *Microbiome* **4**, 42 (2016).
 37. Scheiman, J. *et al.* Meta-omics analysis of elite athletes identifies a performance-enhancing microbe that functions via lactate metabolism. *Nat. Med.* **25**, 1104–1109 (2019).
 38. Capomaccio, S. *et al.* RNA sequencing of the exercise transcriptome in equine athletes. *PLoS One* **8**, e83504 (2013).
 39. Shave, R., Howatson, G., Dickson, D. & Young, L. Exercise-induced cardiac remodeling: Lessons from humans, horses, and dogs. *Vet. Sci.* **4**, 9 (2017).

Reviewers' comments:

Reviewer #1 (Remarks to the Author):

The authors have satisfactorily addressed my concerns. I feel that the manuscript has been greatly improved. Well done.

The only remaining concern I have prior to acceptance for publication is:

Replace "Cluster 1 mainly included the upregulation of many poorly-characterized taxa" with "Cluster 1 mainly included the enrichment of many poorly-characterized taxa".

Reviewer #2 (Remarks to the Author):

I really want to thank the reviewers for carefully considering each of my comments. I believe they have addressed the concerns raised and provided substantiated responses to each. As stated before, I believe authors present an incredible source of scholarly work and data analyses and excellency in the research of the horse gut microbiome, so my comments are meant to be helpful.

The title change now reflects the true nature and scope of the study and I really liked the way the authors have addressed it; however, I still have some minor concerns that seek to help to not overselling the data.

For instance, the abstract (lines 34-36) states that "functionally diverse microbiomes likely increased substrate availability and mitochondrial activity in different tissues, including the heart", but the data still does not prove that nor provide any evidence that the mitochondria or heart benefit from functions or metabolites derived from functionally diverse microbiomes. I know the authors use the word "likely" as an open question, but sentences like this are more fit for the discussion where authors are more free to safely speculate about the possible implications of their data observations.

The main issue here is the absence of phenotypic data (metabolites) that specifically link microbial function with energetic output. While it is true that microbial function denotes greater capacity for glycan degradation, there is no data directly showing that this increased microbial capacity results in more energy supply- along these lines, the host data should then be considered as an association, rather than a consequence of microbial function. This associative nature of the manuscript needs to be highlighted, especially in sections such as L429-451.

I still believe the AMR data is distracting and does not contribute in any way to the scope of this paper. If authors deem this as an interesting finding, this can perfectly be a separate/ additional manuscript.

Last, I suggested a specific "Study limitations" section in my previous comments. I believe this section could be intentional, and would really clarify, for the reader (both specialists and non-microbiome specialists), the extent to which the data can be of value to understand how the equine microbiome contributes to health and performance. This section could also point to research questions to look at/address next and techniques to address these pending questions. This section could include: a) the fact that the data is still associative in nature (e.g. lack of direct evidence that microbial function impacts skeletal/mitochondrial/cardiac function) ; b) care in interpreting the conclusions based on only n=3 high performer horses (e.g. need to be validated across more horses and other populations); and c) any technical limitations (lack of negative controls for shotgun data, and limited scope of metagenomics to infer actual function).

Apart from this, I congratulate the authors for a very nice (exemplary) way of presenting and

summarizing highly multidimensional data. I have been following their work for a while and they are no doubt one of the world leading labs in equine microbiome research.

Reviewer #3 (Remarks to the Author):

My previous comments have been satisfied.

Reviewers' comments:

Reviewer #1 (Remarks to the Author):

1. The authors have satisfactorily addressed my concerns. I feel that the manuscript has been greatly improved. Well done.

AU: We thank the reviewer again for taking the time and effort to review our work and for the lovely comments on our article.

2. The only remaining concern I have prior to acceptance for publication is: Replace “Cluster 1 mainly included the upregulation of many poorly-characterized taxa” with “Cluster 1 mainly included the enrichment of many poorly-characterized taxa”.

AU: We have replaced “upregulation” with “enrichment” in the revised manuscript.

Reviewer #2 (Remarks to the Author):

1. I really want to thank the reviewers for carefully considering each of my comments. I believe they have addressed the concerns raised and provided substantiated responses to each. As stated before, I believe authors present an incredible source of scholarly work and data analyses and excellency in the research of the horse gut microbiome, so my comments are meant to be helpful.

The title change now reflects the true nature and scope of the study and I really liked the way the authors have addressed it; however, I still have some minor concerns that seek to help to not overselling the data.

AU: We would like to thank you for taking the time and effort to revise our manuscript again. We sincerely appreciate all your valuable comments, suggestions, and “criticisms,” which help us improve the quality of the manuscripts. We have now reformatted the manuscript thoroughly by considering your concerns that seek to help not oversell the data.

2. For instance, the abstract (lines 34-36) states that “functionally diverse microbiomes likely increased substrate availability and mitochondrial activity in different tissues, including the heart”, but the data still does not prove that nor provide any evidence that the mitochondria or heart benefit from functions or metabolites derived from functionally diverse microbiomes. I know the authors use the word “likely” as an open question, but sentences like this are more fit for the discussion where authors are more free to safely speculate about the possible implications of their data observations.

AU: The reviewer is correct. We cannot prove that the mitochondria or heart benefit from functions or metabolites derived from functionally diverse microbiomes. We have now revised the paragraph in the abstract to show, however, that we found an association between the functionally diverse microbiomes, circulating plasma metabolites, and mitochondrial-related genes that may have implications for energy production. Thus, the gut microbiome in fit athletes possibly interfered with the metabolism of bioenergetic substrates such as glucose and lipids. This point is addressed in the manuscript in the following way: “Conversely, more complex and functionally diverse microbiomes were associated with higher glucose concentrations and reduced accumulation of long-chain acylcarnitines and non-esterified fatty

acids in plasma, suggesting increased β -oxidation capacity in the mitochondria. In line with this hypothesis, more fit athletes showed upregulation of mitochondrial-related genes involved in energy metabolism, biogenesis, and Ca^{2+} cytosolic transport, all of which are necessary to improve aerobic work power, spare glycogen usage, and enhance cardiovascular capacity”.

3. The main issue here is the absence of phenotypic data (metabolites) that specifically link microbial function with energetic output. While it is true that microbial function denotes greater capacity for glycan degradation, there is no data directly showing that this increased microbial capacity results in more energy supply- along these lines, the host data should then be considered as an association, rather than a consequence of microbial function. This associative nature of the manuscript needs to be highlighted, especially in sections such as L429-451.

AU: We believe that glucose levels, a precursor for glycogen in muscle, the long-chain acylcarnitines, and NEFA concentrations in plasma act as a link between microbial function and the host energetic output. We have rewritten this section to highlight this associative nature as follows: “Conceptually, the higher availability of goods likely influenced the mitochondria function and readouts. This hypothesis paralleled increased glucose availability in serum, a precursor for glycogen in muscle, alongside reduced circulating acylcarnitines and NEFA levels. The acylcarnitine and NEFA drop was possibly not due to reduced substrate availability but to an increase in transport into the mitochondria through the carnitine shuttle, typically for β -oxidation. In concert, mitochondrial-related genes involved in energy resilience, biogenesis, and Ca^{2+} cytosolic transport were simultaneously upregulated in participants with greater cardiovascular capacity suggesting enhanced mitochondrial oxidative phosphorylation and FAO, the two metabolic pathways central to energy production ⁶³. Therefore, bidirectional regulatory circuits between host and inter-microbial components can seemingly increase mitochondrial substrate availability to meet the high energy needs during exertion while optimizing cardiovascular capacity, longer running times, and fatigue resistance.”

4. I still believe the AMR data is distracting and does not contribute in any way to the scope of this paper. If authors deem this as an interesting finding, this can perfectly be a separate/ additional manuscript.

AU: Concerning this point, we agree that ^1H NMR data is not contributing much to the scope of this paper. However, circulating glucose levels (detected by ^1H NMR) were associated with higher cardiovascular fitness. Given that one of the known mechanisms linking gut microbiota and skeletal muscle (*e.g.*, heart) energetics is the availability and storage of glycogen, this result provides evidence of the potential communication between these two organ systems.

Moreover, we consider that to better mine simple and complex associations within the holobiome we need coordinated targeted (acylcarnitine profiles and biochemical assays) and untargeted metabolomic analysis (^1H NMR). Consequently, we would like to keep the ^1H NMR data within the manuscript.

That said, I am aware that the ^1H NMR approach only detects those metabolites with high concentrations (*e.g.*, ketone bodies, amino acids, glucose, pyruvate, choline, etc.) while it can't

distinguish whether the molecular signal has a microbial or host origin. This could be addressed in subsequent experimental studies combining ¹H NMR and mass spectrometry (see “limitations of the study” paragraph).

5. Last, I suggested a specific “Study limitations” section in my previous comments. I believe this section could be intentional, and would really clarify, for the reader (both specialists and non-microbiome specialists), the extent to which the data can be of value to understand how the equine microbiome contributes to health and performance. This section could also point to research questions to look at/address next and techniques to address these pending questions. This section could include: a) the fact that the data is still associative in nature (e.g. lack of direct evidence that microbial function impacts skeletal/mitochondrial/cardiac function) ; b) care in interpreting the conclusions based on only n=3 high performer horses (e.g. need to be validated across more horses and other populations); and c) any technical limitations (lack of negative controls for shotgun data, and limited scope of metagenomics to infer actual function).

AU: I am sorry we did not provide a specific “study limitations” section as you recommended in our previous version. Following your suggestions and the editor’s comments, we have included them at the end of the Discussion section. Please, see the paragraph concerning the several limitations of our study: “The present study contains several limitations. First, interactions should be interpreted cautiously, and the associations cannot be considered direct causal effects. Second, due to the difficulty of sampling elite endurance horses after long races, our observations were made from a small sample size on a single race, with a few samples for independent validation. Replicating results in other athletes with continuous long-term measurements will hence be needed to assess the generalizability of these findings. Third, the current study did not include negative control samples to control for DNA contamination before or during shotgun sequencing. Still, our analyses were restricted to the most prevalent and dominant phylotypes to limit inaccuracy in further data interpretation. Last, functional investigations to identify microbial and host metabolites through ¹H NMR and mass spectrometry coupled with longitudinal meta-transcriptomics and metagenomics are needed to improve inference of microorganism’s functionalities and support the herein reported findings.”

6. Apart from this, I congratulate the authors for a very nice (exemplary) way of presenting and summarizing highly multidimensional data. I have been following their work for a while and they are no doubt one of the world leading labs in equine microbiome research.

AU: I take this opportunity to thank you again for your nice words (I feel honored) and the effort and expertise invested in our article. You have made a significant and positive contribution to the peer-review process.